# Hybrid Molecules Containing Naphthoquinone and Quinolinedione Scaffolds as Antineoplastic Agents

**DOI:** 10.3390/molecules27154948

**Published:** 2022-08-03

**Authors:** Ines Mancini, Jacopo Vigna, Denise Sighel, Andrea Defant

**Affiliations:** 1Laboratory of Bioorganic Chemistry, Department of Physics, University of Trento, 38123 Trento, Italy; jacopo.vigna@unitn.it (J.V.); andrea.defant@unitn.it (A.D.); 2Laboratory of Translational Genomics, Department of Cellular, Computational and Integrative Biology (CIBIO), University of Trento, 38123 Trento, Italy; denise.sighel@unitn.it

**Keywords:** hybrid molecules, multitarget compounds, naphthoquinones, quinolinequinones, isoquinolinediones, drug design, organic synthesis, anticancer, cytotoxicity, medicinal chemistry

## Abstract

In recent decades, molecular hybridization has proven to be an efficient tool for obtaining new synthetic molecules to treat different diseases. Based on the core idea of covalently combining at least two pharmacophore fragments present in different drugs and/or bioactive molecules, the new hybrids have shown advantages when compared with the compounds of origin. Hybridization could be successfully applied to anticancer drug discovery, where efforts are underway to develop novel therapeutics which are safer and more effective than those currently in use. Molecules presenting naphthoquinone moieties are involved in redox processes and in other molecular mechanisms affecting cancer cells. Naphthoquinones have been shown to inhibit cancer cell growth and are considered privileged structures and useful templates in the design of hybrids. The present work aims at summarizing the current knowledge on antitumor hybrids built using 1,4- and 1,2-naphthoquinone (present in natural compounds as lawsone, napabucasin, plumbagin, lapachol, α-lapachone, and β -lapachone), and the related quinolone- and isoquinolinedione scaffolds reported in the literature up to 2021. In detail, the design and synthetic approaches adopted to produce the reported compounds are highlighted, the structural fragments considered in hybridization and their biological activities are described, and the structure–activity relationships and the computational analyses applied are underlined.

## 1. Introduction

Cancer is a complex disease representing the second leading cause of death worldwide, accounting for about 10 million deaths in 2020, as reported by the World Health Organization [1]. Prevention by reduction of risk factors, early detection, and proper treatment are the primary weapons against this disease. Suitable treatments include surgery, radiation therapy, and chemotherapy.

Most anticancer drugs are small molecules able to block the signal transduction pathways for tumor growth and proliferation [2], mainly acting as inhibitors of intracellular protein receptors or as DNA intercalating cytotoxic agents. These small molecules usually address specific molecular targets for the selective elimination of cancer cells. In spite of the efforts made in previous decades, there is still an urgent need to develop novel agents, able to overcome both the resistance and severe side effects associated with currently available drugs. In order to overcome limitations and challenges, a number of multi- and transdisciplinary strategies based on structural biology (crystallographic, cryo-EM, NMR analyses), computational chemistry (molecular docking, dynamic simulations), and medicinal chemistry have been applied.

Within medicinal chemistry, bioisosterism has proved to be efficient in the structural modification of a lead molecule to optimize pharmacokinetic and pharmacodynamic properties. In fragment-based drug discovery, biophysical and biochemical methods are applied to detect the chemical fragments that bind to a specific target, and that are then used to generate drug leads [3]. Furthermore, molecular hybridization has proven to be a valuable approach to obtaining novel potential drugs for the treatment of a number of diseases, including cancer [4].

The pharmacophore unit of a bioactive small molecule is defined as a combination of the molecular features able to ensure optimal interactions with a macromolecular target triggering its biological activity. Molecular hybridization is based on the core idea of combining at least two pharmacophore fragments from different bioactive molecules (such as approved drugs and natural and synthetic compounds) to produce a new single chemical entity. Hybrid molecules are obtained by connecting the pharmacophore scaffolds of the different compounds by directly linking them through covalent bonds, or via a linker. The molecular size obtained, larger than that of origin, often contributes to increasing lipophilicity. For this reason, the linker structure is properly selected to favor the water solubility of the final hybrid compound.

Hybrid molecules can be subjected to structure-activity relationship (SAR) studies, which can lead to the selection of the analogue endowed with the highest specificity, the strongest therapeutic potential, the lowest undesirable side effects and enhanced pharmacokinetic properties. These features are often superior to those of the molecules of single origin or their combination [5].

Hybrids can be designed either to act on the same tumor target by merging two pharmacophore groups from different drugs acting through the same mechanism of action or to act on different tumor targets simultaneously by merging pharmacophores of drugs with different mechanisms of action. In the latter case, the simultaneous presence of the two pharmacologically active entities acting in the same cancer cell results in improved therapeutic efficacy of the hybrid molecule in comparison with the single-target inhibitors. Indeed, the use of a cocktail of two single drugs, which may not have the same efficiency in reaching the action site, is not always that effective. Moreover, by providing a potential synergy among the mechanisms of action involved, hybrid compounds often present an increased effect compared to their parent drugs, allowing a reduction of the administered dose. In addition, it was noted that hybrid compounds overcome most pharmacokinetic drawbacks, toxicity, and side effects of current conventional drugs, and decrease the risk of developing resistance in cancer cells [6,7]. Molecular hybridization is the term that best represents the purpose of this merging strategy and is more commonly used. However, as reported by Decker [8], other terms are also used to define the same approach in drug design, such as “chimera”, “bivalent compound”, “designed multiple compounds”, or multi-target-directed ligand”.

From an availability point of view, the synthetic approach to hybrids is not particularly complex and their access on a large-scale is possible without particular restrictions. Hybridization has been effectively recognized in the last few decades as a promising tool to design new therapeutic molecules with an increased chance of success. Based on all these considerations, it is evident the significant role that hybrids can play addressing the challenges of developing new drugs.

Hybridization has been efficiently applied to anticancer drug discovery, focusing on a broad series of biological targets and the results are widely reviewed [5,9,10,11,12,13]. As recently reported by Shalini et al. [12], twenty-six hybrid compounds have been approved in the last decade as anticancer drugs and sixteen others are involved in clinical trials. These data quantify the success of the hybridization approach in the current development of cancer chemotherapy.

This review presents rational approaches to the design and synthesis, biological data, SAR studies, and computational analyses reported for the anticancer hybrid molecules obtained from or characterized by naphthoquinone (NQ) and related quinolinedione (QD) moieties. After a brief introduction to the context in which the topic and the aim of this review are placed, insights are provided on (i) the role and features of molecular hybridization applied to anticancer agents and (ii) the structural and pharmacophoric relevance of NQ and QD as privileged scaffolds in the mechanisms of anticancer activity. Both topics are discussed, citing review articles. The last part is a detailed collection that focuses on the relevant aspects of each reported work. These works are summarized in suitable figures, highlighting the different scaffolds employed in the structures of the synthetic hybrids. Only those compounds that showed the best inhibition on the indicated human cancer cell lines are numbered.

## 2. Bibliographic Research Methodology

This overview covers publications written in English and reported in peer-reviewed journals up to 2021. The literature reviewed in this work was searched in the PubMed [14], Chemical Abstracts online (SciFinder Scholar) and Google scholar databases, using the following combined three sets of keywords “hybrid/hybridization”, “naphthoquinone/quinolinedione/1,4-naphthoquinone/1,2-naphthoquinone/quinolinedione/isoquinolinedione”, and “antitumor/anticancer/antineoplastic/cytotoxic”.

## 3. Naphthoquinone and Quinolinedione Scaffolds in Antitumor Hybrid Molecules

In the hybrid drug approach, the pharmacophore moieties are selected from approved drugs, or biologically active molecules of both synthetic and natural origin. In their detailed overview dated 2014, Nepali et al. [15] highlighted SAR results and mechanistic insights for anticancer hybrids, classified according to the molecular classes involved. This efficient compilation includes hybrids based on various structural moieties from bioactive natural products. In fact, nature-inspired structures are valuable templates in the production of anticancer hybrids, due to their molecular diversity and biological activities [16].

The naphthoquinone (NQ) structure is formally derived from naphthalene, with a fully conjugated cyclic dione system. Different regioisomers are possible, depending on the relative position of the carbonyl groups (Figure 1): 1,4-naphthoquinone (*para*-naphthoquinone) is the most stable and the most common, while 1,2-naphthoquinone (*ortho*-quinone) and 2,6-naphthoquinone are less frequent. The NQ unit is widely present in natural products isolated from plants, marine organisms, and microbes. Despite the quinone toxicity *in vivo* [11], NQs represent the pivotal structural motif of many natural (e.g., α- and β-lapachones, menadione, plumbagin, juglone, lawsone, and lapachol) and synthetic products, which show a variety of favorable bioactivities, including antibacterial and antitumor effects [15,16,17,18,19,20,21]. Some of these compounds are already in clinical trials [19]. Besides, vitamin K3 (=2-methy-1,4-naphthoquinone) and vitamin K2, which presents a NQ structure bearing a polyprenyl chain (Figure 1), have displayed antineoplastic effects. In particular, health effects of 1,2-NQs have been recently reviewed by Soares et al. [22], who have focused on the danger caused by the action of these compounds on a wide number of targets, with mechanisms not yet fully understood in different cells and tissues, but also on the potential tool to improve their application in the development of new anticancer drugs.

Due to their peculiar structure, NQs are subjected to redox reactions in a reversible manner. In detail, through a semi-quinone species, the benzoquinone is reduced to hydroquinone, which can be converted to the initial form by oxidation (Figure 2). In cells, both 1,4- and 1,2-NQs are easily reduced by different enzymes to semi-quinones and subsequently to naphthalene-diols, increasing the production of reactive oxygen species (ROS). The hydroquinone formed through this mechanism can act as an alkylating agent for DNA, obtaining an irreparable damage leading to apoptosis in cells treated with NQs. Since cancer cells contain higher levels of endogenous ROS and suffer more from oxidative conditions than normal cells, NQs can be exploited to design selective drugs for anticancer strategies [16]. Pavan et al. [23] have efficiently studied the redox mechanism, inducing cytotoxicity, through an NMR-based kinetics in the presence of a reducing agent on a set of juglone-like NQs able to decrease viability of glioma cells.

In addition, the cytotoxicity related to 1,4-NQs is also caused by the fact that nucleophiles such as thiols and amines present in macromolecular targets can react at the C-3 position of the quinone form in these molecules (Figure 1). Moreover, NQs are the substrate of the enzyme NAD(P)H quinone oxidoreductase 1 (NQO1), which activates quinone-like compounds through reduction, is involved in redox process, and, being often overexpressed in cancer tissues, can serve as a target for cancer therapy.

NQs have been shown to be involved in other molecular mechanisms affecting cancer cells, including (i) the inhibition of human DNA topoisomerases I and II (Topo I and Topo II), which are overexpressed in cancer cells and represent a valid target for anticancer agents [16], (ii) the regulation of p53, (iii) the effect shown on the signal transducer and activator of transcription 3 (STAT3) protein, and (iv) the inhibition of EGFR-NF-kB signaling pathway, the last two both linked to inflammation-associated tumorigenesis. Insights on the most relevant mechanisms of action of NQs as anticancer agents have been effectively reported by Pereyra et al. [24]. Detailed mechanisms associated with particular NQ hybrids reviewed in this work will be deepened later in the text.

Quinolinedione (QD, Figure 1) is strictly related to the naphthoquinone structure and is a privileged scaffold for the development of antitumor agents. Indeed, heterocycles play a significant role in the development of anticancer agents, since the additional capacity of heteroatoms to be involved in hydrogen bonds with the target protein can improve selectivity and potency [24]. Among the number of possible isomeric forms, 5,8-quinolinedione (5,8-QD) is the most common, present in many natural and synthetic structures. Biological evaluation on analogues substituted in 6- and 7-position (Figure 1) indicated in most cases that the bioactivity is higher for 6-substituted compounds. However, due to their toxicity, only a few compounds were evaluated in clinical trials. Attempts to design and synthesize analogues endowed with lower toxicity are currently being performed [25].

## 4. Design, Synthesis, and Biological Evaluation of Antitumor Hybrids

### 4.1. Naphthoquinone-Based Molecules

#### 4.1.1. 1,4-Naphthoquinone Scaffold

Several natural 1,4-NQs have shown inhibition of cancer cells and have been considered as models in the design of hybrid molecules. An example is represented by lawsone (=2-hydroxynaphthalene-1,4-dione, Figure 3), which is a metabolite isolated from the leaves of the henna plant *Lawsonia inermis* displaying anticancer activity as Topo II inhibitor. Saluja et al. [26] used it as a moiety in hybridization with urazole scaffolds. A series of hybrid-heterocycles using aldehydes as linkers was synthesized evaluating the ionic liquid 1-butyl-3-methylimidazolium hydrogen sulfate (bmim[HSO_4_]) as an efficient catalyst. Bmim[HSO_4_] could be recovered and used in three subsequent runs without any appreciable loss of efficiency. The one-pot three-component reaction of lawsone and 4-phenylurazole with different aldehydes in the presence of 10 mol% of the ionic liquid provided fifteen new products in high yields. All naphthoquinone–urazole hybrids showed *in vitro* antioxidant activity and some of them inhibited the growth of human breast (T47D), colon (HCT-15), lung (NCI-H522), liver (HepG-2), and ovary (PA-1) cancer cell lines, with compounds **1** and **2** being the most active (Figure 3). More recently, radicals and ions species of these naphthoquinone-urazole hybrids have been evaluated by density functional theory (DFT) method in the gas phase and in water [27].

Gholampour et al. [28] combined the scaffold of the known cytotoxic 2-benzyl lawsone and that of *p*-hydroxyphenylamino-naphthoquinone, based on the greater antiproliferative activity observed for phenyl aminonaphthoquinones compared to alkyl amino analogues. They synthesized ten hybrids with different substituents in *para*-position of the benzyl group. None of them was toxic against HEK-293 normal human kidney cells and compounds **3** and **4** (Figure 3) showed the highest activity against human breast MCF-7 cancer cells, and induced apoptosis. Besides the evaluation of the physicochemical properties and drug-likeness prediction of the synthesized compounds, molecular docking and molecular dynamics allowed to establish the probable interactions between the hybrids and human ATP binding domain of topoisomerase IIα protein.

Lawsone was also considered by Rani et al. [29] as a starting reagent for the ultrasound-accelerated synthesis of 2-(4′-amino-antipyrine)-1,4-naphthaquinone (**5**, Figure 4) by combining it with the commercially available 4-aminoantipyrine, known as a scavenger of reactive oxygen, nitrogen species, and free radicals. The crystal structure of the product was determined by X-ray diffraction method and DFT calculation provided vibrational frequencies. Molecular docking studies on the active site of the kinase CK2 gave score values that have been compared with those of lawsone and FDA approved drugs. Evaluation of proliferation inhibition in MCF–7 human breast and HCT–15 colon cancer cell lines indicated that **5** was more active than lawsone in MCF-7 cell lines. This result is in line with the concept of hybridization as a tool to obtain new molecules displaying improved therapeutic efficacy when compared to the starting inhibitor.

Lawsone, 4,7-dichloroquinoline, cyanuric chloride, and morpholine were used by Fiorot et al. [30] to synthesize three hybrids whose structures were designed based on docking calculation considering the interaction of the lead compound with the phosphoinositide 3-kinase and the 5′ AMP-activated protein kinase. Product **6** resulted to be the most cytotoxic towards human melanoma SKMEL-103 cells, whereas **7** showed no activity (Figure 4).

Structurally defined as biaryl propenones, chalcones are very active against several cancer cell lines, with a mechanism of action involving interaction with tubulin and with its colchicine binding site. A series of naphthoquinone-based chalcone hybrids were obtained by Nguyen et al. [31] via a three-component reaction under microwave irradiation starting from lawsone, *N,N*-dimethyl-formamide dimethyl acetal and acetophenone derivatives (Figure 4). No biological evaluation was performed.

A small library of amino-quinone hybrids was built by Bolognesi et al. [32], combining 1,4-NQ, and the homologue anthraquinone and naphtacenequinone with polyamines, including natural spermidine and spermine and their synthetic variants. It is worth mentioning that a quantitative yield of the products was obtained by the Michael addition using an excess of starting amine, which was then removed by polymer-supported sulfonyl chloride and trisamine resin. According to the MTT assay, all synthetic compounds inhibited the growth of HT29 human colon adenocarcinoma cell line, with the highest activity observed for product **8**, while **9** caused apoptotic EGFR-mediated intracellular signaling (Figure 5). These results highlight the potential multiple activities of polyamine-quinone conjugates, also targeting tumors that have enhanced polyamine transporter (PAT) systems.

Triazole is a chemically stable and biocompatible group, used as amide bioisostere and able to increase the water solubility of a molecule when inserted into its structure. Due to the diverse biological properties shown by triazole-based compounds, including anticancer activity, several research groups synthesized hybrid molecules combining triazole and both 1,4- and 1,2-NQs. By a two-step synthetic sequence involving a copper catalyzed click reaction, Gholampour et al. [33] obtained new 1,4-naphthoquinone-1,2,3-triazole hybrids, designed by the introduction of the triazole unit as a bioisostere of an aromatic amide. ADMET prediction was carried out in comparison with the anticancer drug doxorubicin. All products were evaluated against breast (MCF-7), colorectal adenocarcinoma (HT-29), and leukemia (MOLT-4) cancer cell lines. Most of them displayed cytotoxic activity, with compound **10** (Figure 6) showing the highest activity. Flow cytometric analysis revealed that **10** and two additional compounds arrested cell cycle in the G0/G1 phase in MCF-7 cells, in line with previous observations reported for NQ-based molecules.

Regarding 1,4-NQs, Valença et al. [34] reported the studies on *N*-sulfonyl-1,2,3-triazoles, synthesized using 2-arylamino-1,4-naphthoqionones (series 1) and *N*-propargylated 2-amino-1,4-naphthoquinones (series 2) as precursors. The products were evaluated against eight types of cancer cell lines. The most active compounds were compound **11** belonging to series 1, which also showed no-toxicity to normal cell lines, compound **12** belonging to series 2, and compound **13** among the triazole products. Compound **13** displayed IC_50_ values in the sub-micromolar range in all cancer cell lines tested. The analogues with bromine or a methyl keto group in place of the methoxyl substituent of **13** were also active in the range from 0.31 to 1.20 µM (Figure 6). Of note, the presence of the sulfonyl group in the tosyl unit, which can be subjected to a redox process, is relevant for increasing the bioactivity. The 2-amino analogues **14** and **15** resulted less active, whereas the *N*-sulfonyl triazole derived from lawsone and lapachol, **16** and **17**, respectively, showed submicromolar IC_50_ value. It is noteworthy the comparison of results for **16** and **17**, where replacing the dimethyl allyl unit typical of lapachol with a dimethylvinyl group typical of nor-lapachol structure caused a decrease in the activity on all cell lines evaluated. Doxorubicin and β-lapachone were used as control tests. Besides the antitumor activity of these 1,4–NQ derivatives, the authors reported also on the rhodium-catalyzed conversion of **14** to *N*-substituted conjugated tosyl ene-imine quinones.

Combining the 1,4-NQ scaffold present in menadione and the known pharmacophore 1,2,3-triazole ring, Prasad et al. [35] reported a series of new hybrids (Figure 7), obtained by a three-step procedure involving the copper-catalyzed azide-alkyne cycloaddition. Significant activity was observed for most compounds by evaluation against five cancer cell lines (lung A549, prostate DU-145, cervical Hela, breast MCF-7, and mouse melanoma B-16). Compounds **19** and **20** were the most potent, with **20** showing a higher potency against MCF-7cell line when compared with the standard tamoxifen and menadione of origin. Its activity could be attributed to the induction of cell cycle arrest and apoptosis as displayed by flow cytometric analysis, further confirmed by Hoechst staining, measurement of mitochondrial membrane potential, and Annexin V assay.

Similarly to lapachol, α-lapachone (Figure 7) is a plant metabolite whose antitumor bioactivity is related to the presence of the 1,4-NQ moiety. Structural modifications have been widely considered for the development of potential anticancer drugs, and have been reviewed in 2013 by de Castro et al. [37]. In particular, lapachol and nor-lapachol can be converted to α-lapachone and nor- α-lapachone by acidic treatment. These scaffolds were considered by da Cruz et al. [36] to obtain modified 1,4-NQs showing a triazole unit (nineteen compounds), a sugar moiety (two compounds) with the purpose to increase their bioavailability, an additional quinone group (three compounds) to increase ROS release, a phenyl-thiolated structure to connect chalcogen atoms (Se, S), and arylamino-substituted α-lapachones (eight compounds) (Figure 7). Of note, in the compounds showing arylamino and thioaryl moieties, NH and S act as bioisosteres. Most hybrids resulted to be cytotoxic against eleven cancer cell lines and three normal cell lines, with compound **21** emerging as the most potent. Its effect was similar to that of the analogue presenting a chlorine atom in place of the methoxyl group and to that of α-lapachone and nor-α-lapachone thiophenyl derivatives (**22** and **23**, respectively), which emerged as the most promising molecules considering the selectivity index. Additionally, the authors report on the electrochemical behavior of some of these compounds based on the known redox mechanisms involving naphthoquinones.

Relying on the relevant bioactivities of C-aryl glycosides and on their favorable stability when compared with O- glycosides, which are instead labile to enzymatic and chemical hydrolysis, Chakraborty et al. [38] considered structures containing an NQ aglycone. In particular, the pyranonaphthoquinone structure, present in the *Streptomyces* metabolites griseusin B and mederrhodin A (Figure 8), is responsible for the reductive bioalkylating action, relevant in the mechanism of action against cancer cells. A series of NQ/naphthoquinol–carbohydrate hybrids, obtained in high yield by a regioselective route, were tested against the human cervical cancer cell line, with compound **24** being the most active.

Besides the known anticancer activity of aminoaryl-based 1,4 naphthoquinones (examples of which have already been discussed in some hybridization protocols), Alimohammadi et al. [39] regarded the diaminodiphenyl sulfone structure of dapsone, a synthetic compound with antibacterial and antifungal activities, also used as pharmacophore in the design of anticancer agents. The *N*-methyl piperazine **25** (Figure 9) showed the highest activity against human leukemia cell line, and was shown to be more effective than dapsone against the *S. epidermidis* bacterium. By docking calculation, compound **25** resulted to be a good inhibitor of the BCR-ABL protein, whose tyrosine kinase activity is responsible for cancer cell proliferation in chronic myeloid leukemia. Additionally, **25** showed a strong interaction in the receptor site of T315I Abl mutant, with a free energy value of −11.49 kcal/mol.

Alkannin and shikonin are plant metabolites structurally characterized by the presence of a 5,8-dihydroxy-1,4 naphthoquinone with a six-carbon side chain containing a stereogenic alcohol group presenting (*S*) or (*R*) configuration, respectively (Figure 10). Shikonin is the major component of *Lithospermum erythrorhizon* roots, whose extract has found applications in folk medicine. *In vitro* and *in vivo* evaluation of shikonin including clinical trials was carried out, indicating its potential use as anticancer agent [40]. In detail, shikonin acts by inhibiting tubulin, topoisomerase, and nitric oxide synthase (NOS), finally leading to cell apoptosis. Shikonin-based new molecules have been evaluated to overcome the drawbacks of shikonin related to its poor solubility and strong toxicity. Lin et al. [41] performed a rational hybridization based on the shikonin pharmacophore to design novel dual-action hybrids. The scaffold of the naturally-occurring co-factor α-lipoic acid (= ((*R*)-5-(1,2-dithiolan-3-yl)pentanoic acid), which is involved in multi-enzyme complexes and is known to induce apoptosis in many cancer cells, was considered as the linker to covalently bind the naphthoquinone unit of shikonin to an aromatic ring, reported to affect energy metabolism by inhibiting PDK1 and acting as the second pharmacophore. The computer-assisted drug design method was applied in the design of these tubulin/PDK1 dual inhibitors. In particular, molecular docking was applied for virtual screening, considering substitution on shikonin scaffold, 1,4-naphthoquinone modifications and phenyl substituents. Compound **26** (=(*R*)-1-(5,8-dihydroxy-1,4-dioxo-1,4-dihydronaphthalen-2-yl)-4-methyl pent-3-enyl5-((4R)-2-phenyl-1,3-dithian-4-yl)pentanoate, Figure 10) showed very good energy value for the complex with PDK1, in addition to promoting pyruvate dehydrogenase (PDH) activity, increasing ROS production in tumor cells and causing HeLa cell apoptosis.

Yang et al. [42] applied the hybridization approach to link the 1,4-NQ moiety present in a series of natural products and 4-aza-podophyllotoxin, a structural modification of the natural podophyllotoxin that acts as an inhibitor of the tubulin polymerization, but is too toxic for therapeutic use (Figure 11). The designed hybrids were synthesized by one-pot condensation of 3,4-methylenedioxyaniline, a suitable aldehyde, and 2-hydroxy-1,4-naphthoquinone, through optimized L-proline catalysis. The products were tested on human hepatoma cells (HepG2) and Hela cells, and compound **27** was selected since it displayed a pronounced activity (Figure 11).

In the design of their compounds, Bao et al. [43] started from the evidence that high levels of nitric oxide (NO) and ROS are pro-apoptotic signals in cancerous cells. The hybrids were derived by conjugation through a linker of the natural scaffold of the plumbagin acting as ROS inducer and that of diazeniumdiolates (NONOates) known to be NO donors. A qualitative SAR study involved molecules with different linkers and terminal cycles in the NO donor moiety, which were evaluated for their capability to inhibit the growth of human breast adenocarcinoma (MDA-MB-231), hepatocellular carcinoma (HepG2), pulmonary adenocarcinoma (A549), and colon (HCT-116) cell lines. Compounds **28** and **29** displayed the highest potency (Figure 11), associated with the lowest toxicity on normal cells. Moreover, **28** caused apoptosis in A549 cells in a concentration-dependent manner and was responsible for a ROS generation greater than plumbagin.

Gach et al. [44] planned their hybridization design starting from the properties observed for the naphthofurandione scaffold present in the plant metabolites maturone and avicequinones. This structural motif is associated with the production of intracellular ROS which leads to apoptosis. The phosphoryl group can mimic the tetrahedral intermediates formed in enzymatic reactions involved in the carboxyl group. Further, the known insertion of organophosphorus functionalities into organic molecules can increase their biological activity. Therefore, by combining the naphthofurandione scaffold with the phosphonate moiety, the authors obtained hybrids that were tested against leukemia and breast adenocarcinoma cell lines (Figure 12). 2-(2-Chlorophenyl)-3-diethoxyphosphorylnaphtho [2,3-b]furan-4,9-dione (**30**) was found to be the most active compound, able to generate DNA damage and induce apoptosis. Of note, the cytotoxic activity of **30** is lost by replacing one ethoxyl group with a hydroxyl group, as in compound **31**. Further hybrid molecules containing the 1,4-NQ motif present in substituted 3-diethoxyphosphorylnaphthofurandiones and 3-diethoxyphosphorylbenzoindolediones were investigated, combining them with a phosphonic acid moiety, based on the anticancer activity observed for *N*-phosphonoacetyl-L-aspartic acid (PALA). Concerning this strategy, the same Janecka’s group [45] reported the strongest effect on cell viability of **30** and **31** on hepatocellular carcinoma HepG2 cells (Figure 12). These two selected compounds decreased mRNA expression levels of a series of genes, and induced DNA damage and apoptosis.

The same naphthofurandione scaffold was also considered by Zhou et al. [46], who focused on optimizing the structure of napabucasin (BBI608, Figure 13), a small molecule in phase III clinical trials against some metastatic cancers. Napabucasin is able to inhibit gene transcription driven by the cytoplasmic protein signal transducer and activator of transcription 3 (STAT3), and given the fact that STAT3 is implicated in the main signal pathways able to regulate self-renewal and differentiation of cancer stem cells, it has been reported to affect cancer cell stemness properties. Notably, it has been granted the orphan drug status for the treatment of pancreatic cancer. The aim of the work by Zhou et al. was to develop more potent STAT3 inhibitors able to target cancer stem cells. To this aim, they carried out structure-activity relationship studies on a series of analogues built through hybridization and bioisosterism. In particular, a polar tail was introduced, considering both the imino unit present in the structure of the antineoplastic anthracenyl bishydrazone bisantrene, and amino modifications. Most compounds showed a stronger effect on HepG2 cell growth than napabucasin (IC_50_ = 10.2 µM). Additionally, compounds **32**–**34** (Figure 13) showed a good selectivity index (S.I.), evaluated as the ratio between the IC_50_ values on L-02 cells and those on HepG2 cells, as well as good lipophilicity values according to the predicted LogP and water solubility parameters. Docking calculation with STAT3 protein provided the relevance of additional H bonds involving the groups present on the flexible tail of some compounds. According to the authors’ conclusion, further studies on a large modification space in the side chain are required to increase activity and selectivity.

The design rationale followed by Aly et al. [47] to obtain three series of hybrids considered the thiazole unit present in the cyclin-dependent kinase 1 (CDK1) inhibitors [2.2], the paracyclophane moiety, and a naphthyl unit which in a series of analogues was represented by the 1,4- dihydronaphthoquinone. The synthesized NQ hybrids, which were evaluated for their capability to affect cell viability using the full National Cancer Institute (NCI) 60-cell panel assay, were more active (e.g., compound **35**) than their naphthalene-containing congeners. Compound **36** exhibited a stronger inhibition of melanoma SK-MEL-5 cancer cell growth compared to the reference drug dinaciclib (Figure 14). Moreover, **36** showed a marked downregulation of phospho-Tyr15, an increase in caspase-3, and cell cycle arrest. In addition, it also displayed the highest binding affinity toward the CDK1 according to molecular docking calculations.

In 1997, De Riccardis et al. [48] reported new steroid-anthraquinone hybrids designed starting from the anthracenedione unit of the known anticancer agents doxorubicin and mitoxantrone, and the steroid motif which presents a high ability to penetrate cells and bind to membrane receptors. The synthetic strategy, based on a Diels Alder cyclization to build an internal aromatic ring having NQ as dienophile, provided compounds differing in the presence or absence of an unsaturation in the cholestane chain and in the number of hydroxyl groups in 1,4 positions of the NQ unit. The hybrids were tested *in vitro* at 24, 48, and 72 h on four tumor cell lines, obtaining higher cytotoxicity for the compounds presenting the unsaturated chain, and the highest activity for those having one or two OH groups (e.g., compound **37**, which gave comparable data as doxorubicin) (Figure 15).

In the rationale of their first report on the sugar–oxasteroid–quinone hybrid, Kaliappan and Ravikumar [49] also considered the hybridization of quinones with steroids. In addition, these hybrids also presented a third structural feature represented by a sugar-derived unit. The synthetic strategy to combine the reactivity of the acetal moiety of furanose, the quinone unit and the steroidal backbone involved a sequential intramolecular enyne metathesis, a Diels–Alder cycloaddition and an aromatization reaction. By using this approach, three products with an increased aromatic structure were obtained (Figure 16). More than ten years later, this synthetic approach was applied to obtain a library of sugar–oxasteroid–quinone hybrids with different scaffolds in the sugar derived moiety and in the anthraquinone unit, using six different dienophiles and five different dienes [50]. However, no data on the biological evaluation of these hybrid molecules were reported.

#### 4.1.2. 1,2-Naphthoquinone Scaffold

Despite human health benefits and herbal medicinal uses of many NQs, 1,2-NQ has been associated with severe drawbacks including altered pulmonary function and enhanced inflammatory response. Recently, 1,2-NQ has been reported as a covalent poison of topoisomerase II [51]. Nevertheless, the 1,2-NQ scaffold is present in a series of biologically active molecules, most of them of natural origin. β-Lapachone (Figure 17), which belongs to lapaco-derived metabolites isolated from the bark and heartwood of *Tabebuia* species, is a representative example, evaluated in clinical trials as an inhibitor of pancreatic cancer. It can be easily obtained from lapachol by acid cyclization, according to the procedure generally adopted for synthetic purposes. Another example is represented by naphtho[1,2-b]furan-4,5-dione (NFD), which is the angular analogue of avicequinone B. In detail, the NFD structure contains a fused furan moiety, also fused to the 1,4-NQ scaffold of the natural products avicequinone B and napabucasin, whose biological relevance has already been discussed (Figure 12 and Figure 13, respectively). NFD has been reported to affect cancer cell line viability.

Löcken et al. [52] combined the 1,2-NQ moiety with the 2-acetyl furan unit present in napabucasin to obtain the synthetic, angularly anellated isomer named isonapabucasin (**38**, Figure 17) by a three-step synthetic sequence starting from NFD. Upon *in vitro* evaluation of the compounds of both series of 1,4NQ-derived natural products (lapachol, α-lapachone, avicequinone B, and napabucasin) and of 1,2-NQ derived molecules (NFD, β-lapachone, and isonapabucasin), compound **38** was the most active (Figure 17). Moreover, napabucasin and its isomer **38** inhibited STAT3 phosphorylation, a result also supported by docking calculation, and was proved to affect the redox cycle and generate ROS, in contrast to β-lapachone missing the furan fused moiety.

In the development of new antitumor agents acting on the protein NAD(P)H:quinone oxidoreductase-1 (NQO1), Bian et al. [53] considered the scaffold of the active natural 1,2 -NQs β-lapachone, and tanshinone IIA (TSA), the latter one isolated from *Salvia miltiorrhiza.* The 1,2-NQ unit was merged to the 3-methyl furan ring, eliminating in this way the pyran cycle present in β-lapachone that is hydrolysable to ring-opening products responsible for side effects. The derivatives obtained present a chain with a terminal amine in 2-position. Nineteen analogues were synthesized and their capability to affect NQ1 was evaluated in comparison with β-lapachone and TSA. Compound **39** showed a good IC_50_ value against A549 lung cancer cells, which are rich in NQO1. Moreover, **39** displayed high selectivity against NQO1-deficient H596 cell line (Figure 18). The antitumor activity was due to NQO1-mediated ROS production via redox cycling. Furthermore, docking calculation enabled analysis of interactions between a selected number of synthetic compounds and NQO1.

Li et al. [54] designed, synthesized, and biologically evaluated a new series of β-lapachone analogues. Moreover, they performed molecular docking of some selected molecules in the NQO1 complex. The most promising compound was the benzyl piperazinyl analogue **40**, which is also water soluble under acidic conditions. It inhibited the NQO1-rich A549 cell line (Figure 18) and acted through NQO1-mediated ROS production via redox cycling. Moreover, it showed *in vivo* activity in an A549 tumor xenografts mouse model comparable to that of β-lapachone.

Considering TSA as lead compound, twenty-four new L-shaped *ortho*-quinone analogues were designed and synthesized by Yu et al. [55] The analogues were obtained by removing the C-3 methyl group and by inserting in C-2 position a 1,2,3-triazole substituted by a halogenated or not halogenated aryl unit. A click chemistry approach was applied to produce the triazole by the CuI-catalyzed reaction of the suitable alkyne with azidomethyl-naphthofurandione. The products, in particular compounds **41**–**44** (Figure 18), showed potent cytotoxicity toward prostate cancer (PC3), leukemia (K562), breast cancer (MDA231), lung cancer (A549), and cervical cancer (HeLa) cell lines and some of them induced apoptosis and arrest of the cell cycle. Molecular docking on a selected number of molecules was carried out on the corresponding protein complexes.

Based on the anticancer activity observed for nor-β-lapachone, in 2009 da Silva Júnior, et al. [56] worked on the hybridization of 1,2,3-triazole with 1,2-NQ; notably, they later also considered the 1,4-NQs-trazole hybrids, as reported in Figure 6 [34]. The analogues were synthesized using a click chemistry approach involving the Huisgen reaction with the quinone azide precursor. The products were evaluated against SF-295 (central nervous system), HCT-8 (colon), MDAMB-435 (melanoma), HL-60 (leukaemia), PC-3 (prostate), and B-16 (murine melanoma) cancer cell lines. Compound **45** (Figure 19) displayed the best activity in the melanoma cell line tested, with IC_50_ value lower than those observed for doxorubicin used as positive control. It is known that the electrochemical properties of quinones are relevant for their bioreductive activation. Therefore, the authors carried out electrochemical studies in detail, considering the potential of the first reduction wave (EpIc), which gives a quantitative measure of the reduction ease of an oxidant or electron acceptor. They observed anodic shifts in the reduction potentials, attributable to the ability of the heterocyclic group to affect the voltammetric behavior. However, they did not find a linear correlation between measured reduction potentials and cytotoxicity. Moreover, higher lipophilicity values estimated by the cLogP parameter, were associated with an improved cytotoxicity [56].

Wu et al. [57] synthesized a series of benzotriazole-*nor*-β-lapachone hybrids, designed by molecular docking as NQO1-targeted agents. A good anti-proliferative activity against breast cancer (MCF-7), lung cancer (A549), and hepatocellular carcinoma (HepG2) cells lines was observed for the majority of these products, in line with their NQO1 activity. Compound **46** emerged as the most promising (Figure 19). It was shown to activate ROS production in an NQO1-dependent manner, arrest tumor cell cycle at G0/G1 phase, promote tumor cell apoptosis, and decrease the mitochondrial membrane potential. Moreover, **46** significantly suppressed the tumor growth in a mouse model, with no influences on animal body weights.

1,2-Naphthoquinone-1,2,3-triazole hybrids presenting different substituents on the triazole heterocycle were studied by Chipoline et al. [58]. The synthesis started from the allylation of lawsone, followed by iodocyclization to tetrahydrofuran and further formation of the triazole ring system by the conventional method. The cytotoxic activity of the products was evaluated toward colon (HCT-116) and breast (MCF-7) adenocarcinoma, and the selectivity toward human non-tumor cell line was evaluated on retinal epithelium (RPE) cells. Based on these results, compounds **47** and **48** emerged as the most promising (Figure 19).

The natural benzopyrone coumarin, which displays anti-tumor activity and is involved in a variety of antitumor mechanisms, was used by Martìn-Rodrìguez et al. [59] to be combined with 1,2-NQ to generate the hybrid molecule **49** (Figure 20). In detail, **49** was obtained by an InCl_3_-catalyzed three component reaction of 2-hydroxy-1,4-naphthoquinone, 4-hydroxycoumarin, and 3,4-dimethoxybenzaldehyde under solvent-free conditions. *In vitro* assays proved that **49** is a multi-targeting agent with a submicromolar IC_50_ value for chronic myelogenous leukemia (CML) cell proliferation. Molecular docking considering the kinase catalytic domain of Janus kinase 2 (JAK2) provided values in the range of 8.32–10.1 kcal/mol. The compound emerged as a promising inhibitor of BCR-ABL1-STAT5, an oncogenic signaling pathway in CML.

In recent years, the spirooxindole scaffold has drawn attention due to its presence in the structure of several anticancer compounds, such as the natural metabolite spirotryprostatins A and MI-888 that are in preclinical studies. Adopting the diversity-oriented synthesis approach, Wu et al. [60] produced a series of spirooxindole-O-naphthoquinone-tetrazolo [1,5-a]pyrimidine hybrids, synthesized through a one-pot three-component reaction involving 2-hydroxy-1,4-naphthoquinone, differently substituted isatins, and 5- aminotetrazole in refluxing acetic acid as the optimized condition. These molecules were highly cytotoxic against the cancer human liver cell line HepG2, with a good selectivity ratio related to the normal cell line LO2; the most relevant values were obtained for compound **50** (Figure 20).

For their structural hybridization, Zhou et al. [61] considered the scaffold present in the 2-substituted perimidine that are highly cytotoxic, and the 1,2–NQ scaffold present in the naturally occurring sesquiterpene *ortho-*quinone mansonone F, which displays topoisomerase II inhibition. The authors focused specifically on the development of catalytic Topo II inhibitors, which are superior to Topo II poisons, since the latter ones have shown several side effects in clinical therapy. Most of the compounds synthesized exhibited potent cytotoxicity and compound **51** (Figure 21) displayed IC_50_ values lower than 1 μM against four cancer cell lines and strong potency (0.47 μM) against the HL-60/MX2 cell line, a mitoxantrone resistant cell line derivative of the HL-60 cell line. In addition, compound **51** was a non-intercalative catalytic Topo IIα catalytic inhibitor and could induce the apoptosis of Huh7 cells in a dose-dependent manner.

Considering the 1,2-NQ scaffold present in a number of antitumor molecules including β-lapachone and mansonone F, Wu and Zhang [62] studied a series of hybrids presenting the 1,3,4-thiadiazolo [3,2-a]pyrimidine moiety. The compounds were obtained by a one-pot three component reaction starting from 2-hydroxy-1,4-naphthoquinone, the suitable aldehyde and the appropriately 5-substituted-2-amino-1,3,4-thiadiazole (Figure 21). Since this synthetic approach could also lead to the production of the isomeric product containing the 1,4-NQ unit, the structure of the desired compounds was carefully established by 2D-NMR analysis, supported by heteronuclear multiple bond correlation (HMBC) experiments. All molecules showed a good cytotoxicity against human colon (HCT116) and liver (HepG2) cell lines as compared to taxol taken as the control, with compound **52** as the most active. Moreover, the majority of these compounds were less cytotoxic toward non-cancerous L02 cells.

Organoselenium compounds have been widely studied for their biological properties including antitumor activity, deriving from their antioxidant properties and ability to mimic the activity of some selenoenzymes. By a significant and complete study, Vieira et al. [63] have reported structures containing the β-lapachone scaffold functionalized with both alkyl and aryl selenium and sulfur moieties. These chalcogen-β-lapachone hybrids were obtained by a rapid, ecocompatible, and efficient microwave irradiation of lapachol with the suitable nucleophiles and dimethylsulfoxide oxidant, catalyzed by iodine. The products, structurally characterized also by X-ray analysis, were evaluated against a wide series of cancer cell lines, including leukemia, human colon carcinoma, prostate, human metastatic prostate, ovarian, central nervous system, and breast. Comparing the IC_50_ values on specific cell lines for compounds presenting either a selenium or a sulfur atom (compounds **53**, **54** and **55**, Figure 22), it is evident that selenium analogues displayed the highest activity. In detail, compounds **55**, **56,** and **57** were found generally to be the most potent. Furthermore, compound **58** emerged as the most promising against HL-60, displaying good selectivity on non-tumor cells (human peripheral blood mononuclear cells (PBMC), two murine fibroblast lines, and canine kidney epithelial cells), with a selectivity index higher than that observed for doxorubicin related to PBMC. The possible mechanism involving these compounds having two redox centers has been also investigated.

#### 4.1.3. 1,4- and 1,2-Naphthoquinone Scaffolds

As previously discussed in this review [33,37,56], the molecular hybridization involving quinone and triazole moieties has been proven to be an important strategy to prepare new bioactive molecules, as it involves an organoselenium unit [64].

In the period 2011–2018, da Silva Júnior’s group focused on hybrids of 1,2- and 1,4-NQs with these structural features and obtained an in-depth view of structure-activity correlation. In this regard, Bahia et al. [64] considered hybrid compounds containing 1,4- and 1,5-disubstituted-1,2,3-triazoles and quinonoid units, the latter ones including both the 1,2-NQ unit present in the structures of β-lapachone and nor-β-lapachone and the 1,4-NQ unit that characterizes α-lapachone and nor-α-lapachone (Figure 23). Moreover, the presence of an iodine atom on the triazole unit allows building of C–C bonds for producing further derivatives. *Ortho*-quinone compounds were produced starting from natural lapachol to obtain the 3-azide nor-β-lapachone, which was subsequently used in a catalyzed click reaction with an alkyne substrate to give a product presenting the 1,5-disubstituted regiochemistry as confirmed by X-ray crystallographic analysis. A similar procedure was adopted to synthesize the 1,4-quinone products. Compounds **59** and **60** showed trypanocidal activity and **60** was also more active than the precursor β-lapachone, against several cancer cell lines. The authors remarked that these preliminary data are of interest for further investigations, also considering that these compounds are accessible through viable synthetic routes.

More recently, Costa et al. [65] reported another series of NQ-hybrids presenting the 1,2,3-triazole scaffold functionalized by an aryl moiety with different substituents. The synthesis involved the reaction between 2-hydroxy-1,4-naphthoquinone and vinyl 1,2,3-triazoles presenting the suitable aryl substituents, with cerium (IV) ammonium nitrate as the oxidizing agent. The isomeric 1,4- and 1,2-NQ products were obtained with a slight excess of the 1,2-NQ form. The compounds were tested toward three human cancer cell lines (mammary gland MDA-MB231, colon Caco-2, and lung Calu-3) and non-tumor cells (Vero). Compound **61** was the most selective toward Caco-2 cells. Docking calculations supported the blockage of Topo I and IIα as one of the mechanisms of action responsible for the cytotoxic effect of **61** in Caco-2 cells.

Da Silva Júnior et al. [66] synthesized other quinoid compounds and their activity was investigated against leukemia (HL-60), melanoma (MDA-MB435), colon (HCT-8), and central nervous system cancer cell lines. Three series of products can be mainly distinguished (Figure 24). Compound **62** was selected as the most active in the imidazol anthraquinone set of molecules (second series), the 1,2-NQ bromoazide **64** was more potent than the 1,4-NQ bromo azide **63**, and the β-lapachone-based 1,2,3-trazoles **65**–**68** displayed the strongest cytotoxicity activity, with IC_50_ values in the very low µM range.

Hybrid structures built on the nor-β-lapachone scaffold, containing the frequently used 1,2,3-triazole unit and an additional 1,4-NQ unit, were reported by Jardim et al. [67]. Selected for their intrinsic capability to generate ROS, these molecules resulted very active against four cancer cell lines, with IC_50_ values lower than 2 µM (i.e., **69**, Figure 25) but they did not show selectivity when related to normal PBMC. A second series of compounds derived by combining nor-β-lapachone and chalcone scaffolds, were synthesized using amino-functionalized chalcones with different aryl substituents. All products were highly active (i.e., **70**) and among them the methyl analogue **71** emerged as the most promising based on its good selectivity for the HL-60 cell line, with a selectivity index even better than that observed for doxorubicin. Lastly, α-lapachone derivatives were found to be inactive (IC_50_ ˃ 10 µM).

The strategy adopted by da Cruz et al. [68] to design novel bioactive hybrids was based on the antitumor activity displayed by compounds having (i) a 1,2,3-triazole or a substituted phenylamino unit on the C6 or C5 cyclic ether present in α-lapachone,β-lapachone and nor-β-lapachone and (ii) a chalcogen atom. The authors expanded the chemical diversity of the previously studied hybrids by adding a selenium-containing unit, which could act as a second redox centre, in addition to the quinone system. Figure 26 reports the main structural features of the new 18 selenium-containing quinone-based 1,2,3-triazoles, obtained by a copper-catalyzed click chemistry approach using a phenylselenium-functionalized azide. Of note, the production of enantiomerically pure molecules **72** and **73** was accessible through a sequence involving a chiral squaramide catalyst.

These new and already reported compounds, providing a total of forty-nine hybrids, were evaluated against six human cancer cell lines (acute promyelocytic leukemia HL-60, colon HCT-116, prostate PC3, glioblastoma SF295, melanoma MDA-MB-435, and ovarian OVCAR-8). Figure 27 reports a selection of the most active compounds. Some of them were found to be more potent than β-lapachone or doxorubicin and displayed a very good selectivity versus normal cells. Additionally, it was found that the most active molecules were specifically bioactivated by NAD(P)H:quinone oxidoreductase 1 (NQO1), which contained glutathione peroxidase (GPx)-like activity and were able to induce apoptosis associated with ROS production (**74** and **75**). Other interesting compounds were **76** and **77**, showing IC_50_ values in the sub-µM range, associated with a very good selective index for **77** (Figure 27).

More recently, the effect of chalcogen atom (Se, S) on activity was investigated by considering aryl groups with different substituents, or by replacing the phenyl ring with 2-thiophenyl or alkyl chains [69]. A first series of 1,4-NQ and aryl-Se hybrids was obtained by rhodium-catalyzed C-H bond activation and click reactions (Figure 28). It was observed that the reaction yield of 1,2,3-triazole formation was strongly affected by the substituents on the aromatic ring of the selenium alkynes. The cytotoxicity of all products was evaluated against five human cancer cell lines and also correlated to activity in non-tumor L929 cells. Compounds **78**–**80** emerged as the most active, with **78** displaying the higher selectivity ratio related to normal cells.

A further contribution to the structural diversity of NQ hybrids was given by the work performed by Gontijo et al. [70]. They obtained novel compounds by coupling the NQ units of α- and β-lapachone to the fluorescent BODIPY (=boron-dipyrromethene) core [70]. The synthetic strategy involved the use of the click chemistry of BODIPY-alkyne with the suitable azide on the NQ units. By evaluation on cancer and normal cell lines, compound **81** was identified as the most interesting (Figure 29). Studies on mechanisms involving lipid peroxidation and determination of reduced and oxidized glutathione were performed, indicating that the activity of **81** may be related to ROS production. Additionally, analysis by fluorescence confocal microscopy indicated that the cytotoxic compound **82** was preferentially localized in the lysosomes of cancer cells.

### 4.2. Quinolinedione and Isoquinolinedione-Based Molecules

#### 4.2.1. Quinoline-5,8-dione Scaffold

As a structural modification of 1,4-NQ, quinolinedione, and specifically 5,8-QD, are moieties responsible for anticancer and other biological activities shown by natural and synthetic products. The alkaloid streptonigrin isolated from *Streptomyces* displaying anti-leukemic properties, and the ascidian metabolites ascidiathiazones A and B are examples of molecules belonging to this class. This moiety is considered promising in the synthesis of bioactive agents [25]. However, to date few studies have been reported considering this scaffold in the synthesis of anticancer hybrids.

Very recently, Janeka and coworkers have expanded their previous studies on antitumor naphthofurandione–phosphonate hybrids [44,45] from which compound **30** emerged as the most active (Figure 12). By combining the quinolinedionefurandione scaffold with the phosphonate moiety, they obtained a series of new diethoxyphosphorylfuroquinoline-4,9-diones hybrids (Figure 30) [71]. Starting from 6,7-dichloroquinoline-5,8-dione, the two *N*,O-*syn* and *N*,O-*anti* regioisomers were produced and their stereochemistry was assigned by ^13^C- and ^31^P-NMR analyses. By evaluation on promyelocytic leukemia (HL-60) and breast cancer adenocarcinoma (MCF-7) cell lines, and on the non-tumor human umbilical vein endothelial cells (HUVEC) and mammary gland/breast (MCF-10 A) cell lines, several compounds were found to be highly cytotoxic with IC_50_ values also lower than 0.1 mM. Of note, the *N*,O-*syn* regioisomers were more active than the corresponding *N*,O-*anti* forms, and the quinolinedione analogues were significantly more active than the corresponding naphthoquinones (e.g., **83** compared with **30**). Additionally, a similar mode of action was proved for compounds **83** and **84** that showed the highest selectivity index between cancer and non-tumor cells. In HL-60 cells, these compounds enhanced intracellular ROS generation and NQO1 depletion which led to the cell cycle arrest in the S-phase, reduced cell proliferation, and induced DNA damage and apoptosis.

#### 4.2.2. Quinoline-5,8-dione and 1,4-Naphthoquinone Scaffolds

Diosgenin is a steroidal sapogenin, derived as aglycone by hydrolysis of saponins abundant in the Dioscoreaceae plant family, reported with various biological properties, including antioxidant and antitumor activities. Recently, Li et al. [72] have reported new diosgenin-1,4-quinone hybrids, designed starting from the known modifications of diosgenin structure at C-3 and C-26 positions, which are able to modulate the bioactivities of the corresponding derivatives. A series of compounds were synthesized by introducing in these positions substituted 1,4-NQ and 5,8-QD fragments. The evaluation of these molecules on three human cancer cell lines (MCF-7, HepG2, and HeLa) indicated that the activity is related to the naphthoquinone structure. Compound **85** was 35-fold more cytotoxic against HepG2 liver cell lines compared to diosgenin (Figure 31). This hybrid activated the mitochondrial apoptosis pathway in the HepG2 cell line and molecular docking showed interaction of its 1,4-NQ unit inside the active site of the NQO1 enzyme.

Kadela-Tomanek et al. [73] reported the hybridization of 1,4-NQ and 5,8-QD unit with betulin, a triterpene isolated from the bark of birch trees which showed anticancer activity. A series of compounds were synthesized by linking 6,7-dichloroquinoline-5,8-dione or its 2-methyl analogue to betulin in three different positions. The products were tested on seven human cancer cell lines (glioblastoma SNB-19, melanoma C-32, and Colo-829, breast MCF-7, T47D, MDA-MB-231, and lung A549), obtaining selective inhibition, with some compounds (e.g., **86** in Figure 32) displaying IC_50_ values in the submicromolar range, and proving to be more potent than the betulin and cis-platin used as controls. According to a qualitative SAR study, the activity depends on the type of naphthoquinone present in the molecule structure, with 1,4-NQ conferring higher activity than the 5,8-quinolinedione unit. The 1,4-quinone interaction of hybrids with the NQO1 enzyme observed by molecular docking supported the observed increase in biological activity. Selected compounds showed a mitochondrial apoptosis pathway in A549 and MCF-7 cell lines.

The introduction of the 1,4-quinone moiety on the betulin scaffold caused an increase in the lipophilicity of these hybrids. In the light of the relevant role of this parameter in the pharmacodynamic and pharmacokinetic properties, the authors investigated the lipophilicity of their compounds by experimental and computational methods. The results indicated no correlation between the biological activity and the lipophilicity of the tested compounds [74].

Recently, the same authors expanded the study to hybrids presenting the betulin scaffold connected by a triazole linker to the 1,4-quinone unit, also including isoquinoline 5,8-dione [75]. The tested compounds displayed a higher activity against cancer cell lines characterized by increased levels of the NQO1 protein. Indeed, these molecules were identified as efficient substrates for NQO1 by enzymatic studies and were shown to activate the mitochondrial apoptosis pathway in the A549 cells. Compound **87** (Figure 32) was one of the most potent, with an IC_50_ value lower than that observed for cis-platin in the A549 cell line. The 1,4-quinone structure promoted the interaction of the hybrids in the active site of the enzyme, as pointed out by docking calculation.

In our study on naphthoquinone and 5,8-quinolinedione-derived antitumor molecules, we have reported new aminoquinone hybrid molecules, designed starting from the structure of PT-262, a synthetic inhibitor of the ROCK1 kinase [76]. This cyclic or functionalized amino scaffold was connected to the timethoxyphenyl fragment, present in several antitumor compounds, such as in the natural combretastatins and colchicine, which act as potent tubulin inhibitors [13], in the podophyllotoxin which acts as both tubulin and Topo II inhibitor and in the synthetic MTB0B214, an apoptotic inductor through caspase 9 activation. In the rational approach to the hybrids, the molecules to be synthesized were selected by docking calculations of the corresponding complexes with ROCK 1, human Topo II and tubulin. Additionally, computational evaluation of their drug likeness was performed and a good bioavailability was predicted. By evaluating the products against the full panel of 60 human cancer cell lines of the National Cancer Institute (NCI), GI_50_ values in the 10 nM-10 μM range were obtained. Compound **88** (Figure 33) was identified as the most active in the series and it was evident that the relative position of the *N,N* heteroatoms was crucial in affecting selectivity and bioactivity, e.g., the *N,N*-*anti* isomer **89** was found to be almost always more active than *N,N***-***syn* form.

#### 4.2.3. 5,8-Isoquinolinedione Scaffold

As an isomeric structure of 5,8-QD (Figure 1), 5,8-isoquinolinedione is also a moiety present in antitumor molecules and represents an interesting scaffold in the synthesis of antitumor hybrids, although few cases have been reported so far.

Da Silva et al. [77] designed their hybrids considering that the isosteric substitution of one aromatic CH by a nitrogen atom was shown to increase the potency of the corresponding quinone analogues, as evident in the case of the antitumor pixantronedimaleate (BBR 2778) if compared with the related mitoxantrone. Other examples of bioactive isoquinoline-5,8-dione compounds are the antitumor marine metabolites caulibugulones A–D (Figure 34). A carbohydrate was involved as the second structural moiety involved in the hybridization, based on the anticancer strategy to target the Warburg effect, which describes the ability of cancerous tissues to consume larger amounts of glucose in comparison to normal tissues. The 5,8-isoquinolinedione-carbohydrate hybrids were synthesized by the 1,4-addition reaction between 5,8-dioxo-5,8-dihydroisoquinoline and different aminocarbohydrates, to produce a series of molecules with and without the carbohydrate units and/or the halogen atom, which have been then subjected to a SAR study. The activity of the hybrids was investigated against lung (H1299) and prostate (DU-145) human cancer cells and Vero and MDA-MB231 normal cells. Compound **90** was the most active (with cell viability % values of 37.85 and 78.47 at 100 µM corresponding to H1299 and DU-145, respectively), while showing a selectivity index on the normal cells tested higher than that of the reference drug.

#### 4.2.4. 5,8-Isoquinolinedione and 1,4-Naphthoquinone Scaffolds

The work by Mello et al. [78] aimed at identifying a molecule acting as an activator of the 5′ AMP-activated protein kinase, an interesting target for cancer therapy since the critical role played in regulating cell metabolism, growth, and survival. Regarding the quinone pharmacophore, both 1,4-NQ and 5,8-isoquinolinedione were considered. The latter scaffold is present in the structure of FRI-1 (Figure 35), which has been very recently reported as an inhibitor of mitochondrial bioenergetics by redox disruption and an inductor of apoptosis in MCF7 breast cancer cells [79]. Hybrid naphthoquinone-4-oxoquinoline and isoquinoline-5,8-quinone-4-oxoquinoline derivatives were synthesized by a final Michael reaction of the suitable aryl amine to the 1,4 quinone groups. Compound **91** showed the most promising results, being selectively cytotoxic on breast cancer cells in comparison with non-cancer cells (Figure 35), and being able to specifically activate AMPK in cancer cells and to act as inhibitor of mTOR signaling, leading to a strong modulation of metabolism.

## 5. Summary Remarks

Based on the data of the 48 works collected and reported in this overview, it is evident that the most studied NQs are 1,4-NQs (57%), followed by 1,2-NQs (33%), and less frequently by compounds characterized by 5,8-QD (7%) and isoquinoline-5,8-dione scaffold (3%). The structures of lawsone and mainly lapachone and nor-lapachone, characterized by a fused tetrahydropyrane and tetrahydrofurane ring, respectively, have drawn great attention among the natural NQs. On the other hand, by analyzing the other structural fragments used in the design and synthesis of antitumor hybrids, 1,2,3-triazole are the most used (almost 25%), followed by aryl and alkyl amine and fused furan ring, in turn followed by sugar ans steroid, and in lower amount chalcogen (with Se four times more frequently used than S), chalcone, coumarin, and trimethoxyaryl units (Figure 36). In the hybrids presenting a linker, 1,2,3-triazoles were widely applied, because they offer the possibility of covalently binding the two molecular fragments constituting the hybrid through the copper-catalyzed click reaction between the suitable alkyne and azide functionalities. A relatively small number of studies considered chiral molecules, as well as analogues resulting from bioisosterism (e.g., triazole/aromatic amide, NH/S, and NH/O).

Qualitative SAR studies on the series of synthesized hybrid analogues were common to almost all the reviewed works, except for a couple of reports where the synthesized molecules have not been tested. In a number of cases the new hybrids show higher activities than the reference NQs and/or the other molecular scaffolds considered, in line with the aim of molecular hybridization.

About twent works reported inhibition of cellular proliferation in human cancer cell lines with IC_50_ values in the submicromolar range and an even smaller number reported results on selectivity versus normal cell lines. Moreover, only in two cases did the biological evaluation also includes an *in vivo* test (Table 1). Investigations on the mechanism of action were present in a number of cases, the results of which are summarized in Table 2.

Computational approaches were considered in about twenty cases, including DFT calculation, drug-likeness prediction, molecular dynamics (Table 1), and especially molecular docking, to study the interaction of the hybrid in complex with the suitable protein (Table 1 and Table 2). However, it has to be noted that docking calculations were usually applied to the structures of the most promising hybrids resulting from biological evaluation and only in two cases they were used for *in silico* selection of the hybrids to be subsequently synthesized.

## 6. Conclusions

The studies here reviewed show that NQs and the related 5,8-QD are privileged structures in the development of new antitumor agents, due to the wide range of biological processes affected by the molecules containing them, and due to the effective synthetic possibilities which these scaffolds offer for the production of novel hybrids. In a number of works, the compounds derived from molecular hybridization showed a more favorable antitumor activity than the molecules of origin, confirming the validity of this approach. Promising perspectives may arrive from an optimized procedure involving a more efficient correlation among organic synthesis, computational analysis, and experimental biological investigation, allowing the development of new potential drugs to be evaluated in clinical trials.

## Figures and Tables

**Figure 1 molecules-27-04948-f001:**
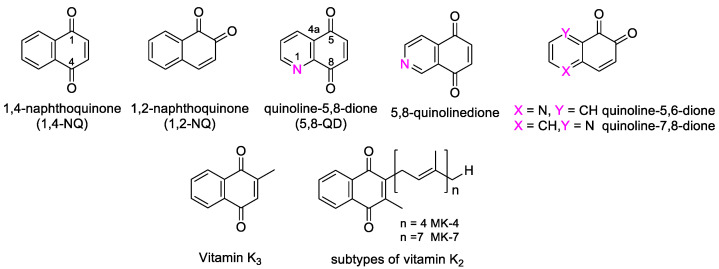
Molecular structures of naphthoquinones and the most common quinolinediones, and of vitamin K2 and K3.

**Figure 2 molecules-27-04948-f002:**
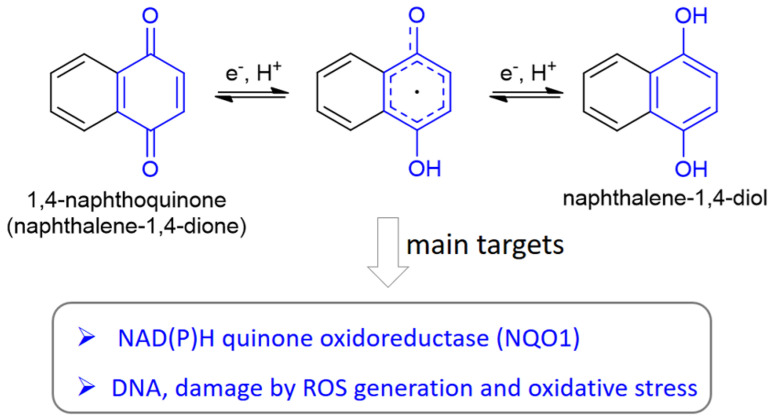
Redox reactivity of naphthoquinones, here shown for 1,4-NQ with the benzoquinone and hydroquinone units depicted in blue, and the main biological targets involved in this chemical reactivity.

**Figure 3 molecules-27-04948-f003:**
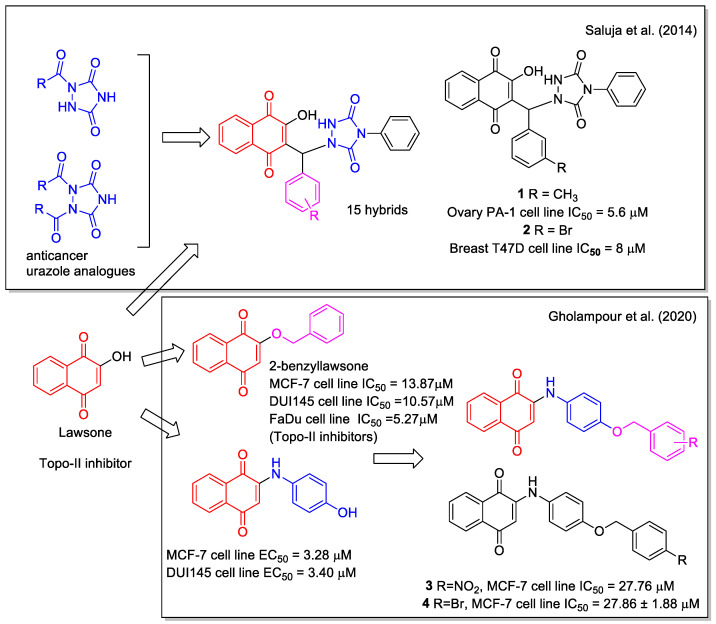
1,4-Naphthoquinone-urazole hybrids as reported by Saluja et al. [26] and arylaminonaphtoquinone hybrids as reported by Gholampour et al. [28].

**Figure 4 molecules-27-04948-f004:**
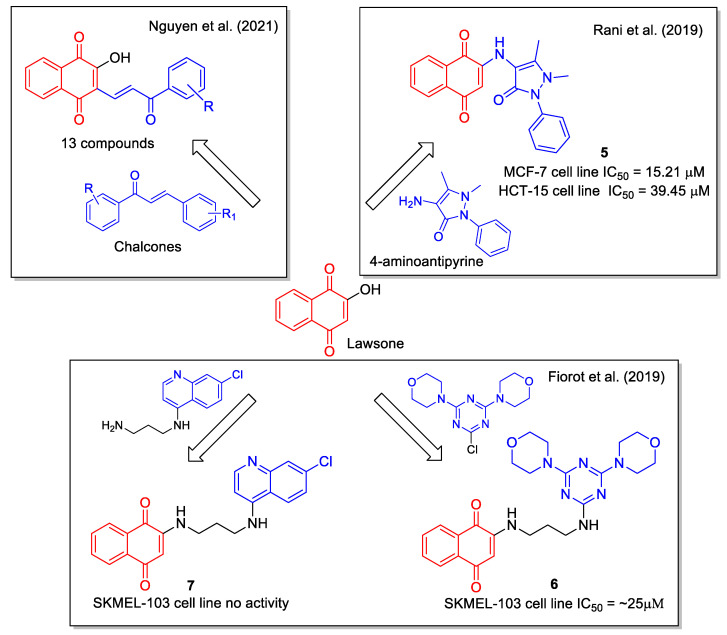
1,4-Naphtoquinone hybrids with aminoantipyridine as reported by Rani et al. [29], with chloroquinoline by Fiorot et al. [30] and with chalcone by Nguyen et al. [31].

**Figure 5 molecules-27-04948-f005:**
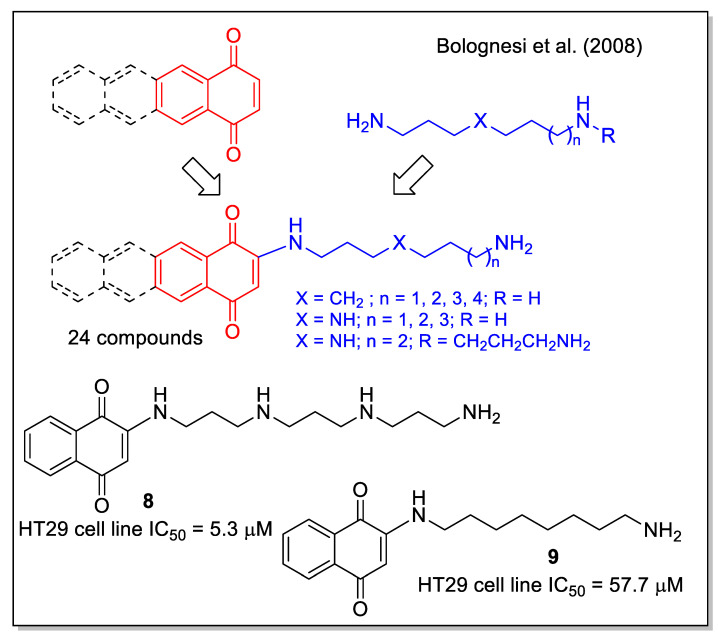
1,4-Naphtoquinone-polyamine hybrids as reported by Bolognesi et al. [32].

**Figure 6 molecules-27-04948-f006:**
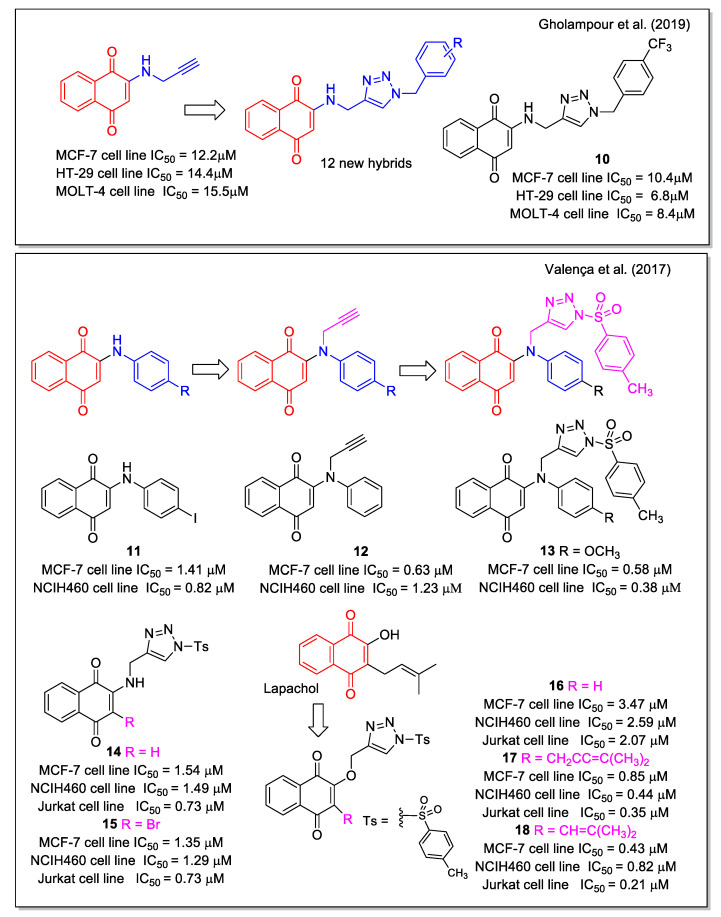
1,4-Naphthoquinone hybrids with 1,2,3-triazole as reported by Gholapour et al. [33] and with *N*-sulfonyl 1,2,3,-triazole by Valença et al. [34].

**Figure 7 molecules-27-04948-f007:**
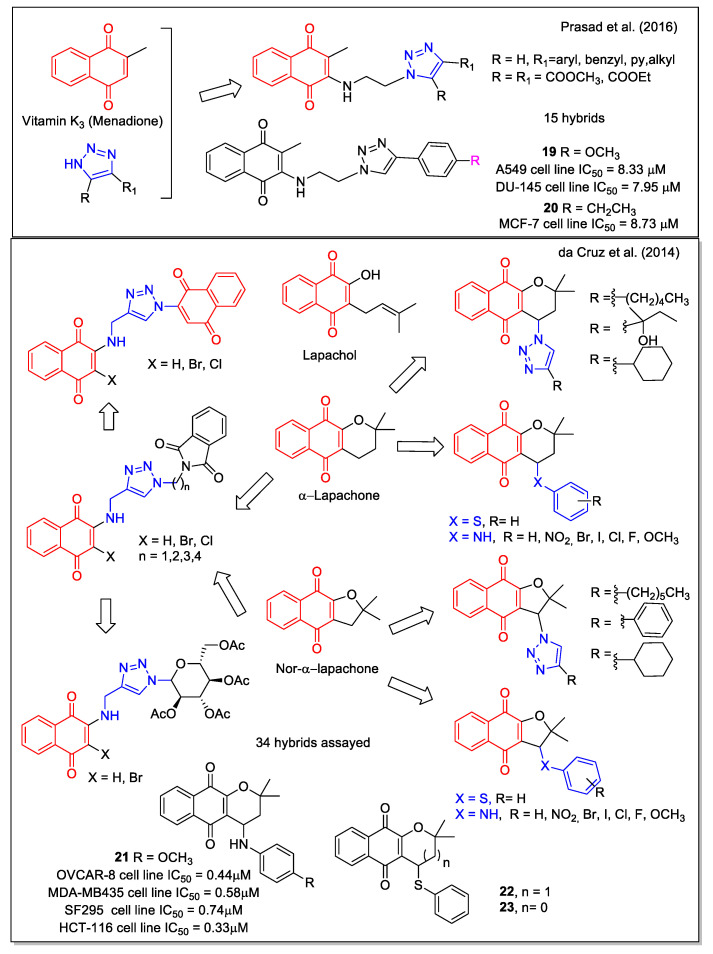
1,4-Naphthoquinone hybrids as reported by Prasad et al. [35] and by da Cruz et al. [36].

**Figure 8 molecules-27-04948-f008:**
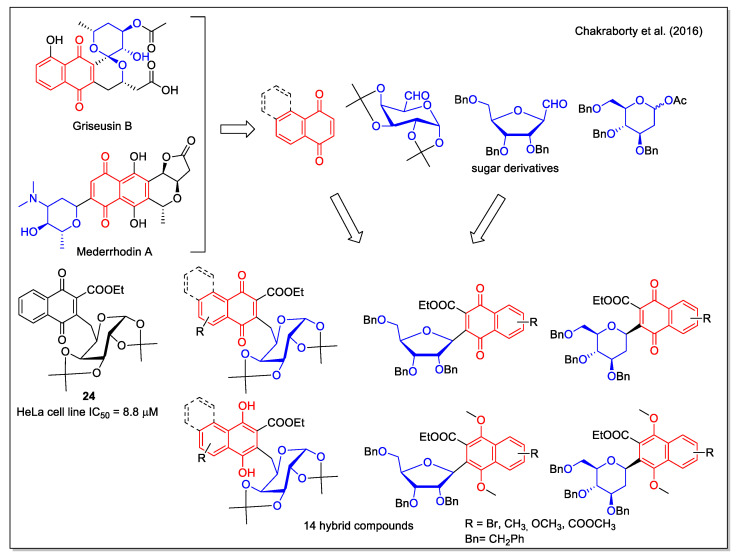
1,4-Naphthoquinone-carbohydrate hybrids as reported by Chakraborty et al. [38].

**Figure 9 molecules-27-04948-f009:**
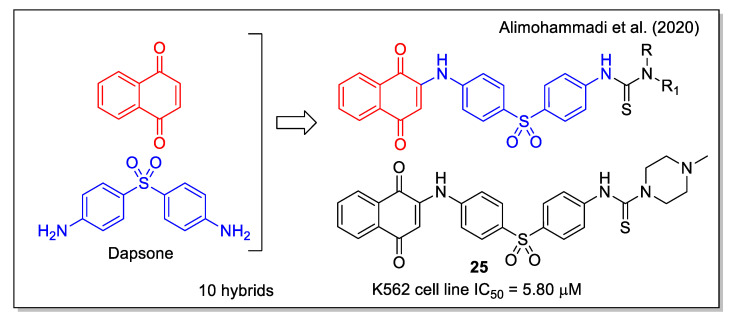
1,4-Naphthoquinone-diaminodiphenyl sulfone hybrids as reported by Alimohammadi et al. [39].

**Figure 10 molecules-27-04948-f010:**
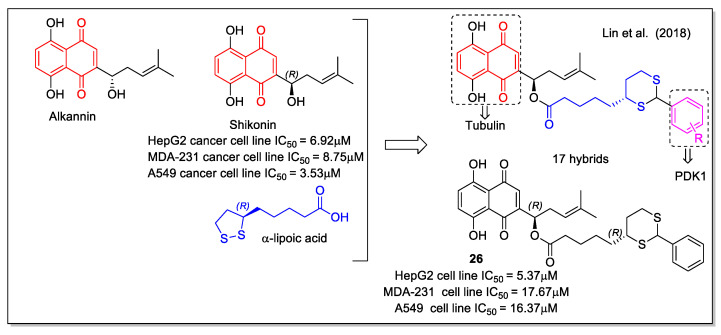
Shikonin-lipoic acid hybrid as reported by Lin et al. [41].

**Figure 11 molecules-27-04948-f011:**
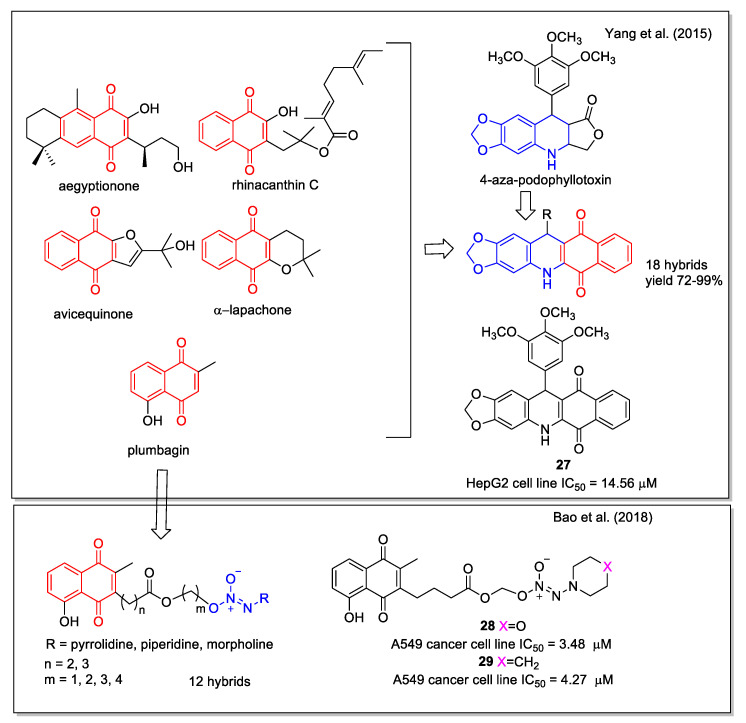
1,4-Naphthoquinone hybrids with 4-aza-podophyllotoxin as reported by Yang et al. [42] and with diazeniumdiolate scaffold by Bao et al. [43].

**Figure 12 molecules-27-04948-f012:**
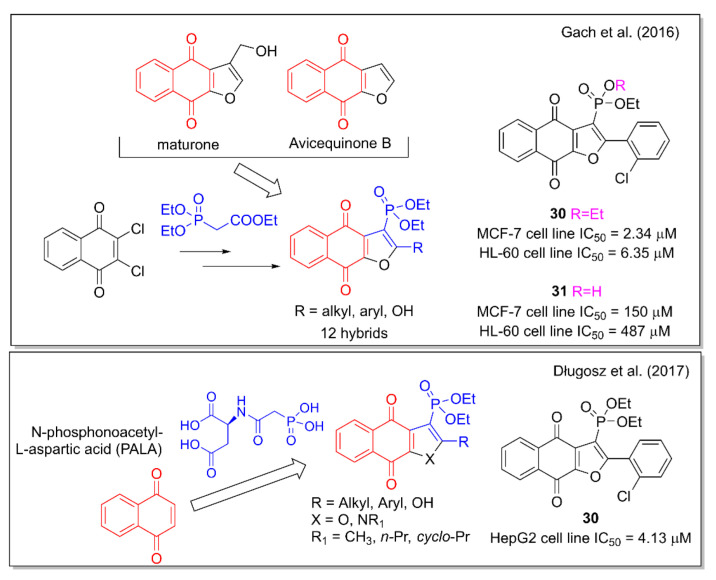
Naphthofurandione–phosphonate hybrids as reported by Janecka and coworkers [44,45].

**Figure 13 molecules-27-04948-f013:**
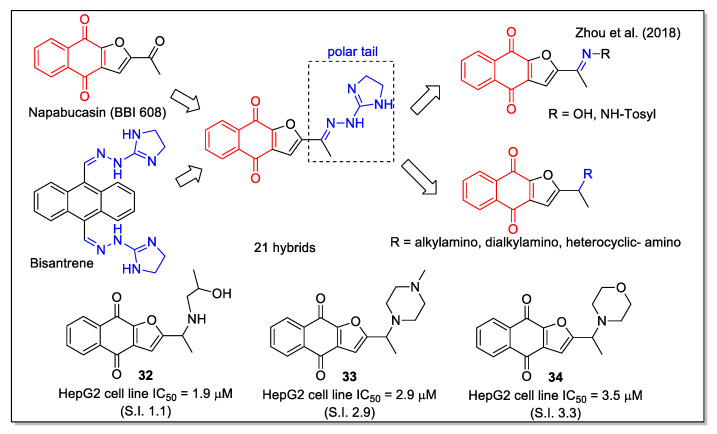
1,4-Naphthoquinone hybrids with imino- and amino-urane scaffold as reported by Zhou et al. [46].

**Figure 14 molecules-27-04948-f014:**
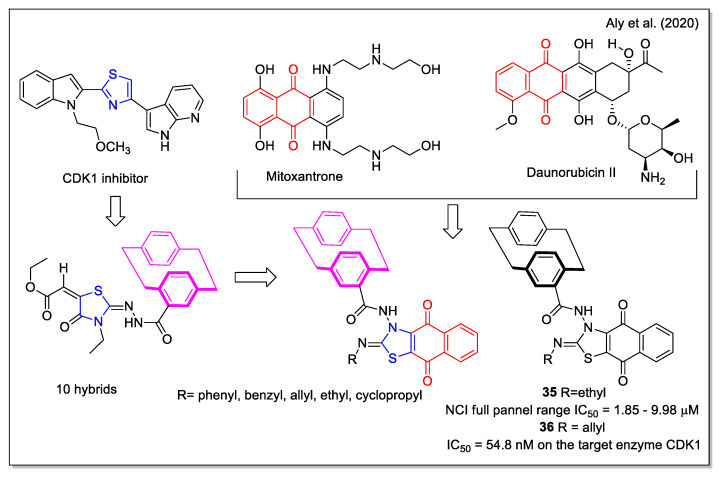
1,4-Naphthoquine-thiazole-paracyclophane hybrids as reported by Aly et al. [47].

**Figure 15 molecules-27-04948-f015:**
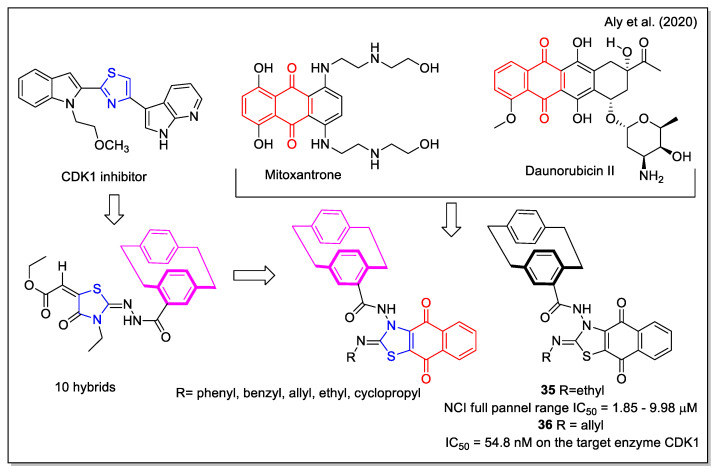
Steroid-anthraquinone hybrids as reported by De Riccardis et al. [48].

**Figure 16 molecules-27-04948-f016:**
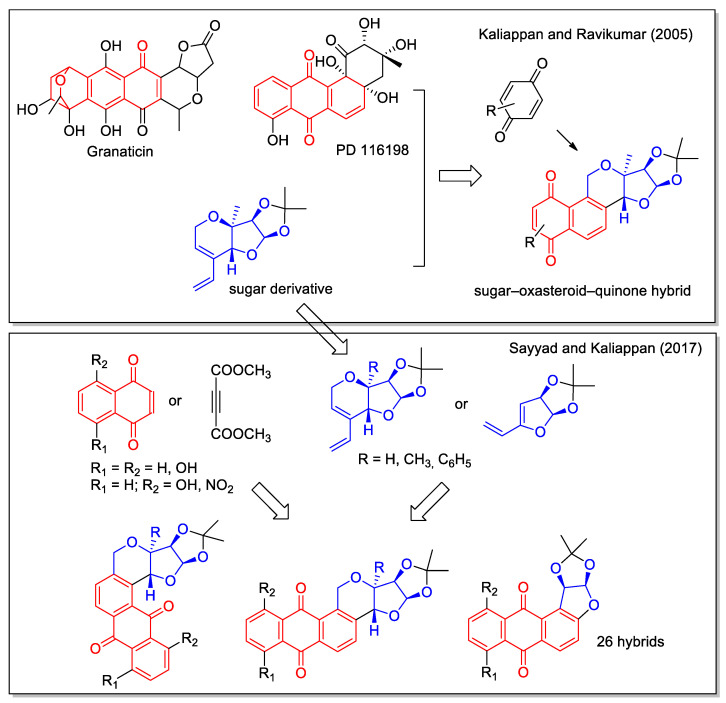
Sugar–oxasteroid–quinone hybrids as reported as reported by Kaliappan and coworkers [49,50].

**Figure 17 molecules-27-04948-f017:**
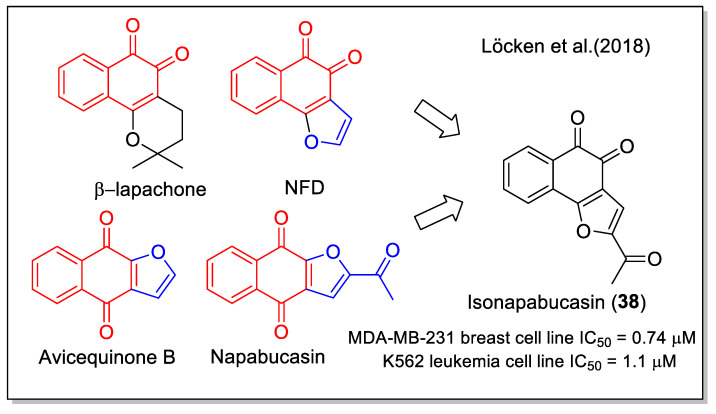
The 1,2-naphthoquinone-2-acetyl furane hybrid as reported by Löcken et al. [52].

**Figure 18 molecules-27-04948-f018:**
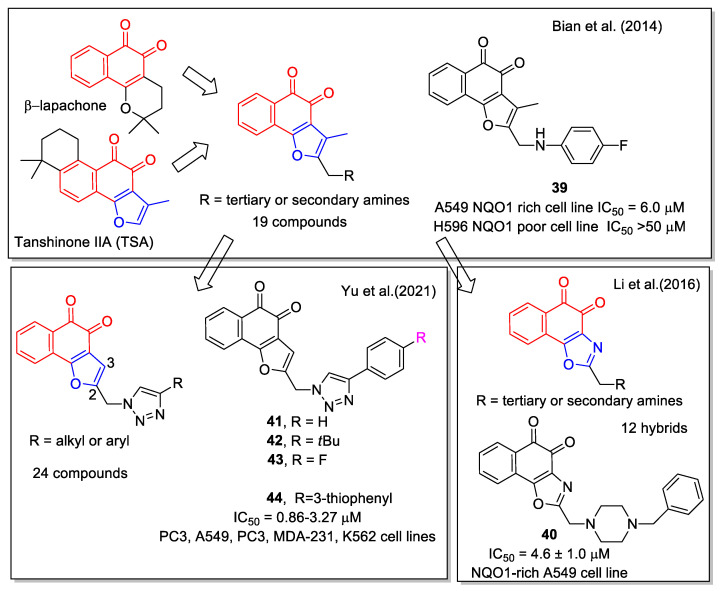
1,2-Naphthoquinone hybrids with 2-functionalized furan as reported by Bian et al. [53] and with 2-amino oxazole by Li et al. [54], and by Yu et al. [55].

**Figure 19 molecules-27-04948-f019:**
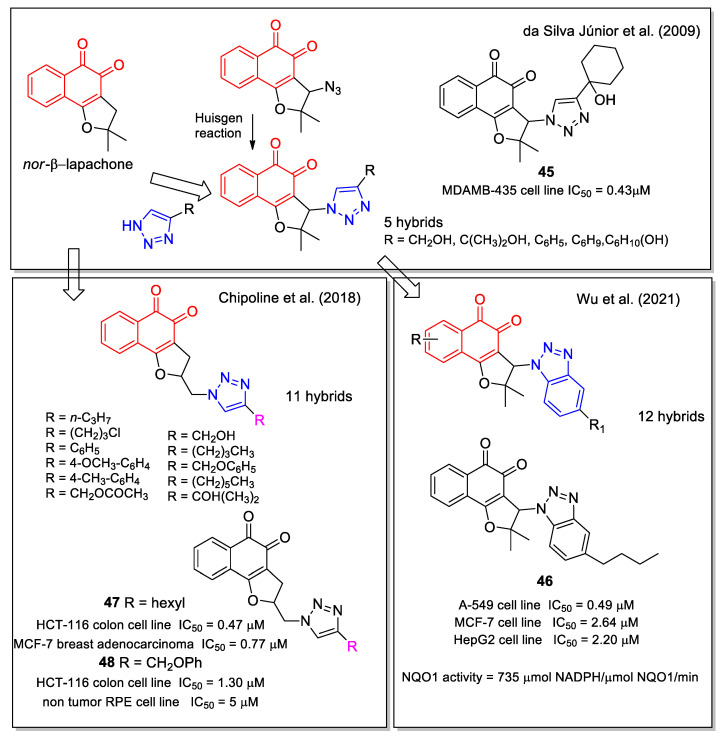
1,2-Naphthoquinone-substituted 1,2,3-triazole hybrids as reported by da Silva Júnior, et al. [56], by Wu et al. [57] and by Chipoline et al. [58].

**Figure 20 molecules-27-04948-f020:**
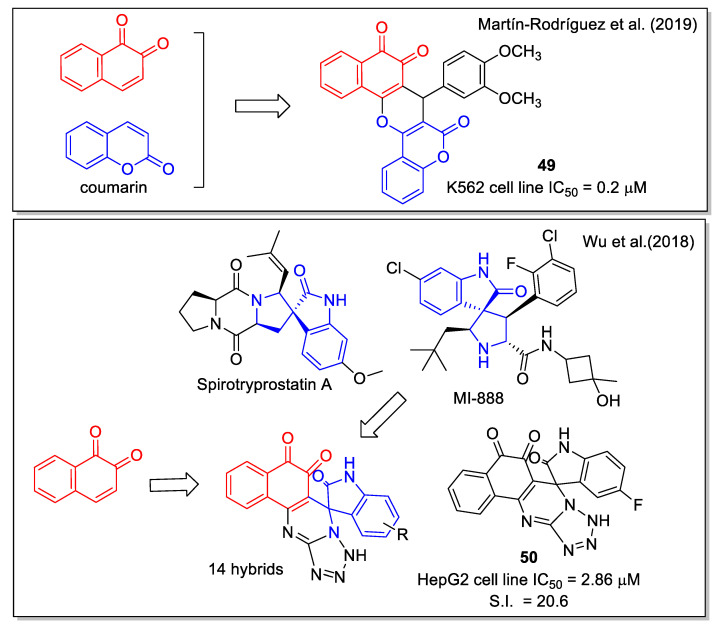
1,2 Naphthoquinone hybrids with coumarin-dimethosxyaryl scaffolds as reported by Martin-Rodriguez et al. [59] and with spirooxindole-tetrazolopyrimidine by Wu et al. [60].

**Figure 21 molecules-27-04948-f021:**
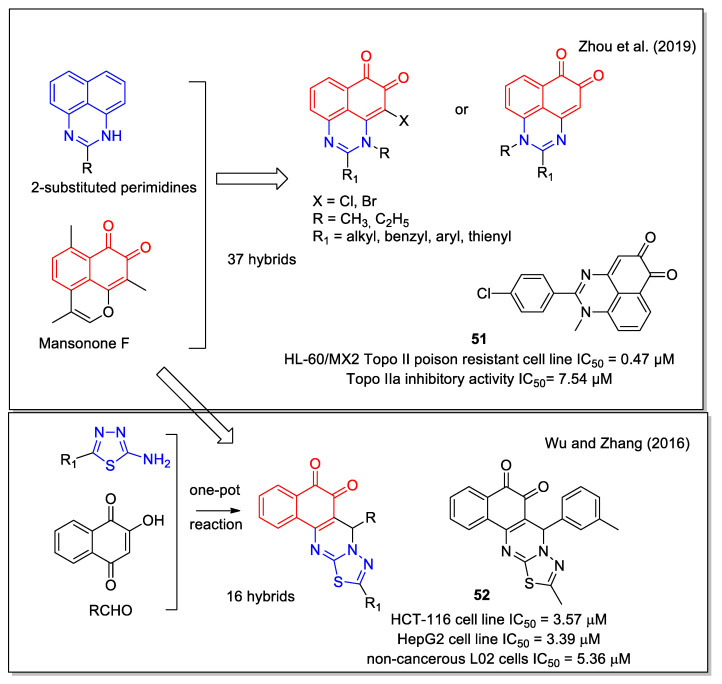
1,2-Naphthoquinone hybrids containing the 2-substituted pyrimidine scaffold as reported by Zhou et al. [61] and the 1,3,4-thiadiazolo [3,2-*a*]pyrimidine scaffold by Wu and Zhang [62].

**Figure 22 molecules-27-04948-f022:**
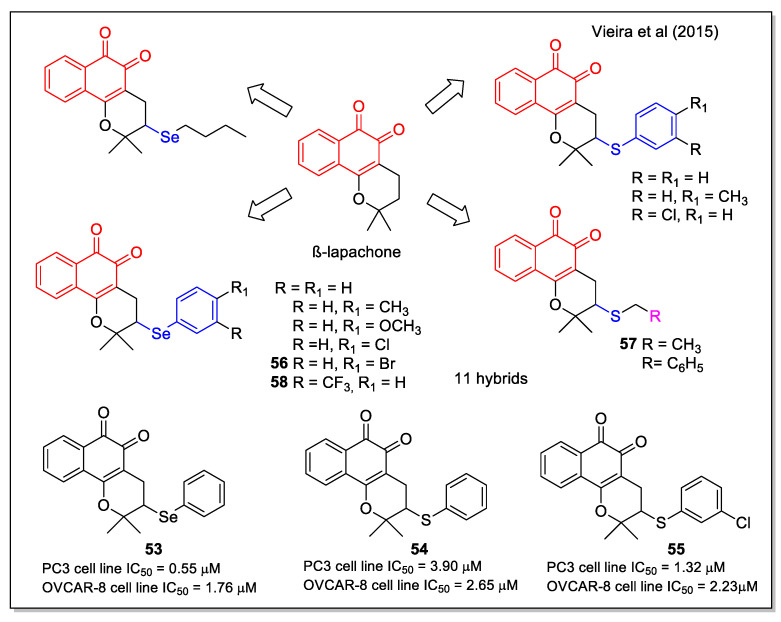
β-Lapachone-chalcogen (Se or S) hybrids as reported as reported by Vieira et al. [63].

**Figure 23 molecules-27-04948-f023:**
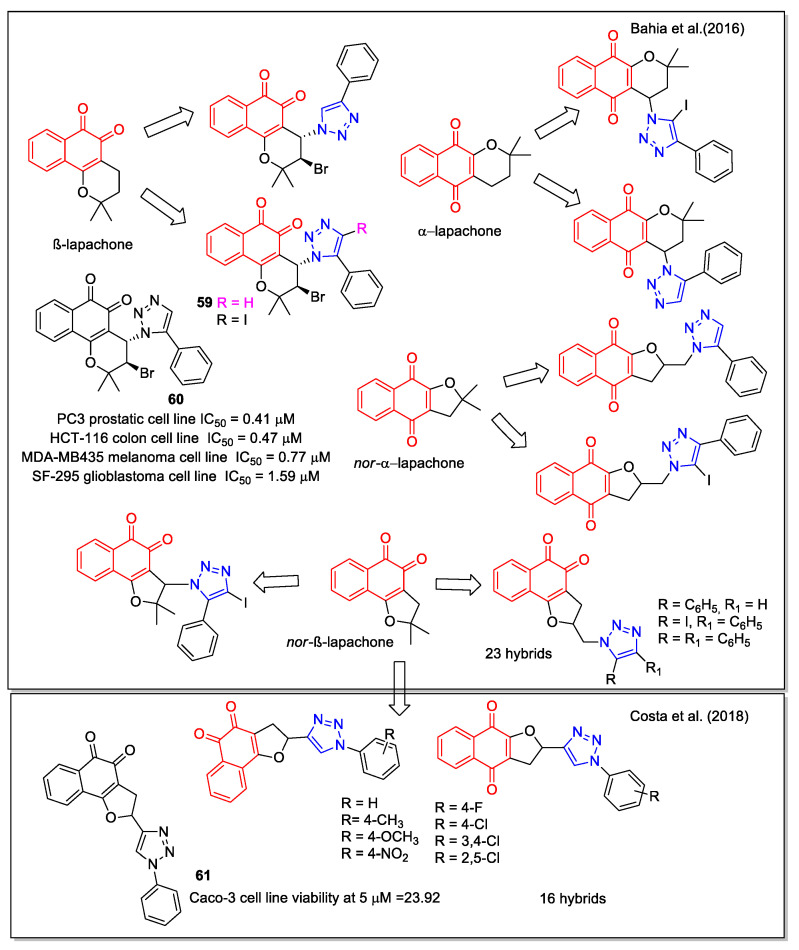
Lapachone or nor-lapachone–substituted 1,2,3-triazole hybrids as reported by Bahia et al. [64], and by Costa et al. [65].

**Figure 24 molecules-27-04948-f024:**
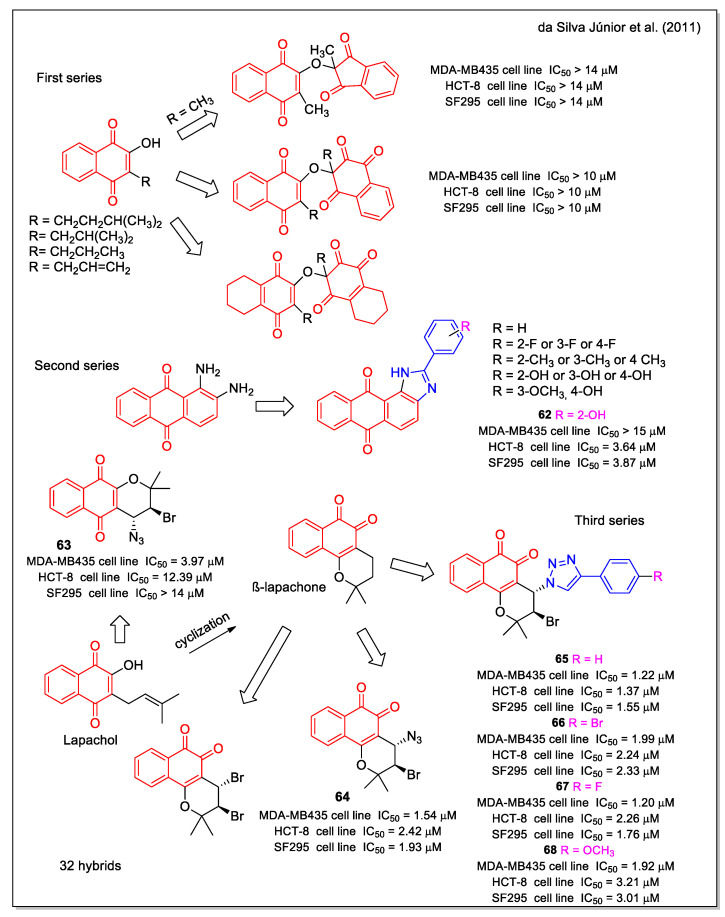
1,4- and 1,2 -Naphtoquinone hybrids as reported by da Silva Júnior et al. [66].

**Figure 25 molecules-27-04948-f025:**
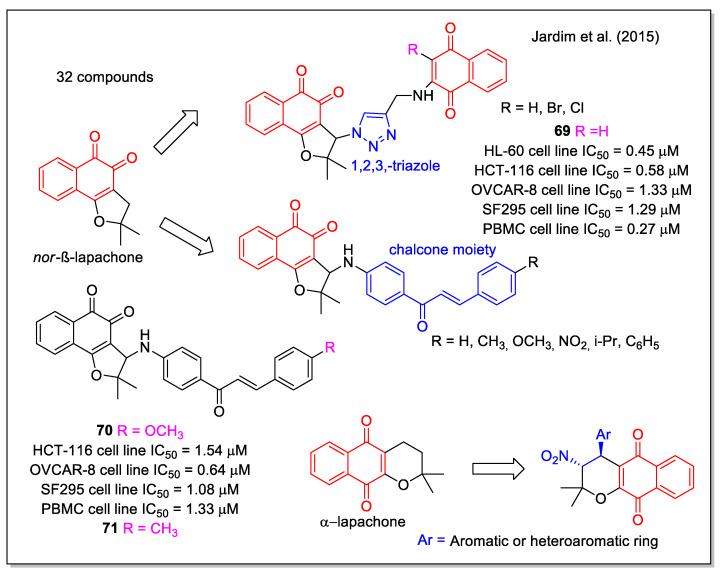
Lapachone hybrids with 1,2,3-triazole or chalchone, or nitro group units as reported by Jardim et al. [67].

**Figure 26 molecules-27-04948-f026:**
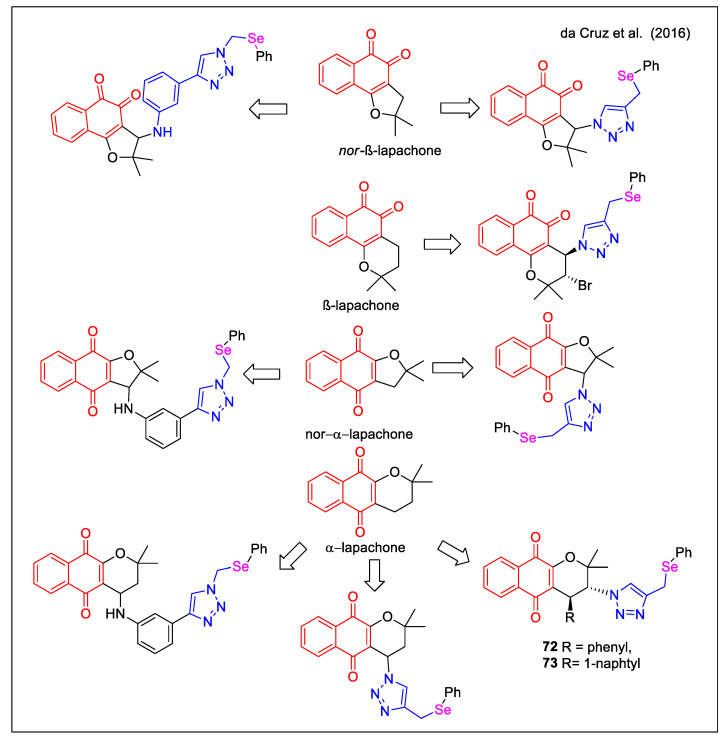
Lapachone-1,2,3-triazole with selenium functionalization hybrids as reported by da Cruz et al. [68].

**Figure 27 molecules-27-04948-f027:**
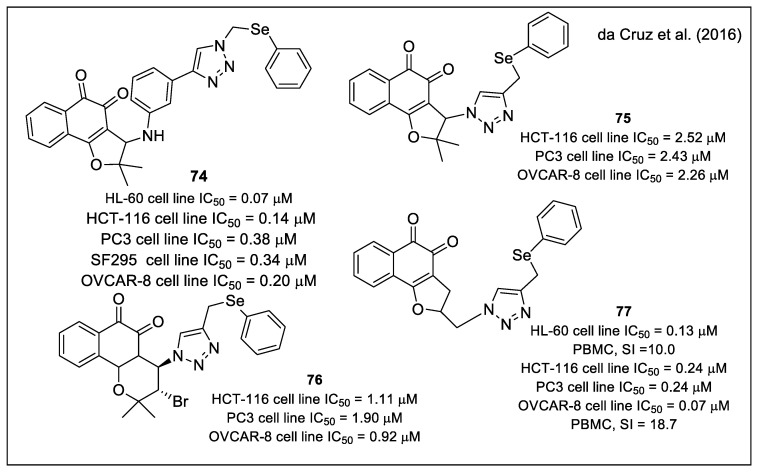
Molecular structures and cytotoxic data of the most active Se- containing compounds as reported by da Cruz et al. [68].

**Figure 28 molecules-27-04948-f028:**
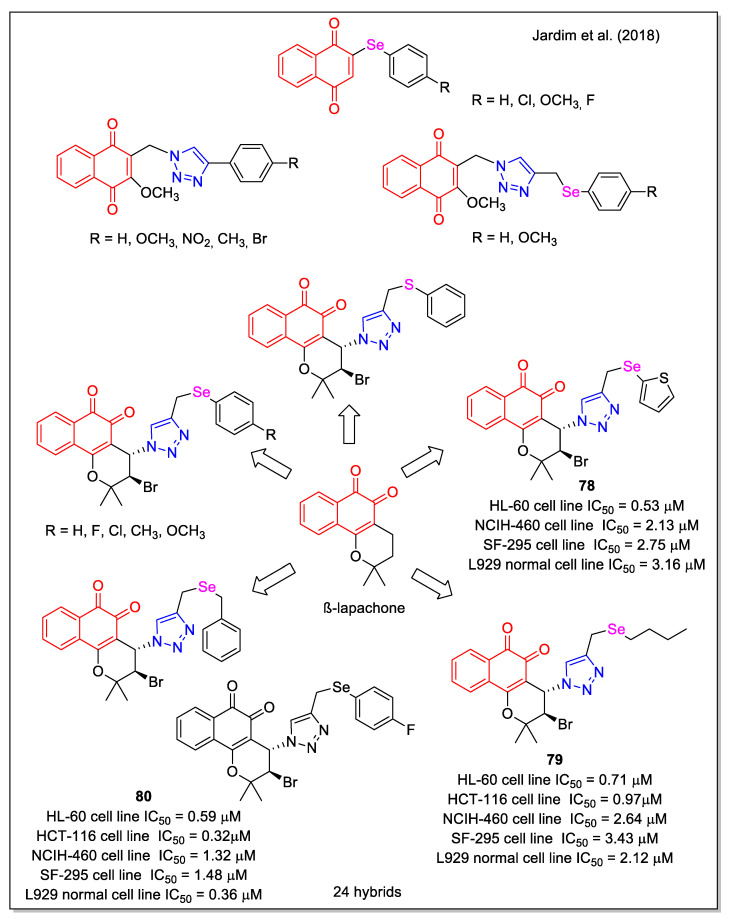
1,4- and 1,2-Naphthoquinone–organoselenium hybrids as reported by Jardim et al. [69].

**Figure 29 molecules-27-04948-f029:**
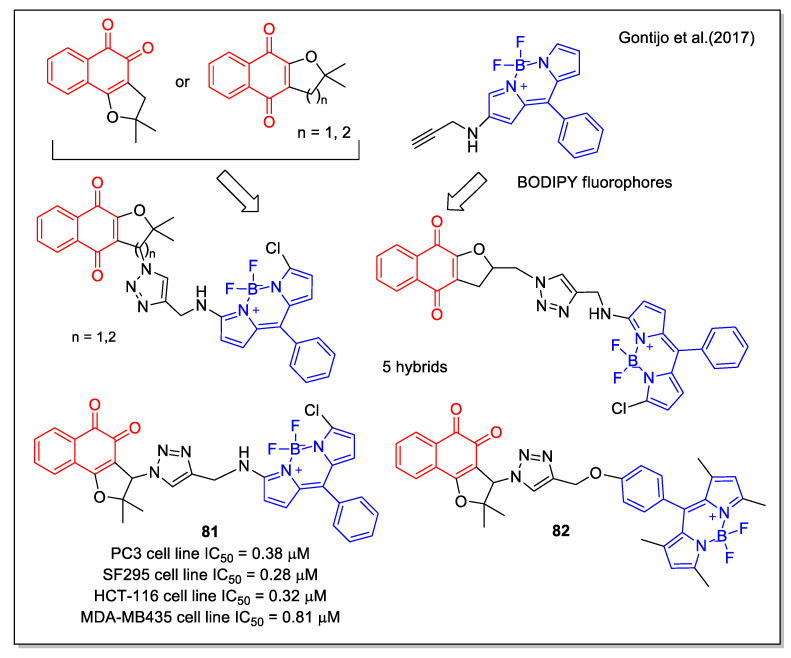
Nor-lapachone-BODYPY fluorophore hybrids as reported by Gontijo et al. [70].

**Figure 30 molecules-27-04948-f030:**
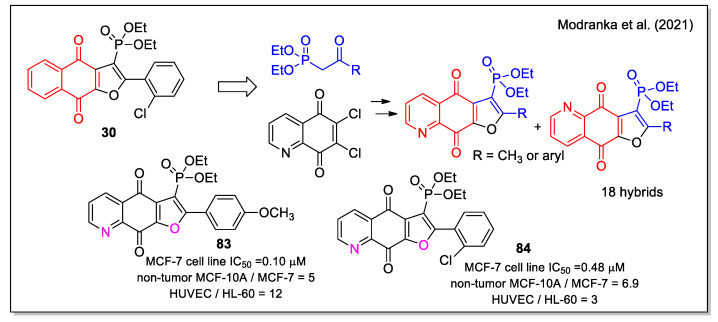
Quinolinedionefurandione–phosphonate hybrids as reported by Modranka et al. [71].

**Figure 31 molecules-27-04948-f031:**
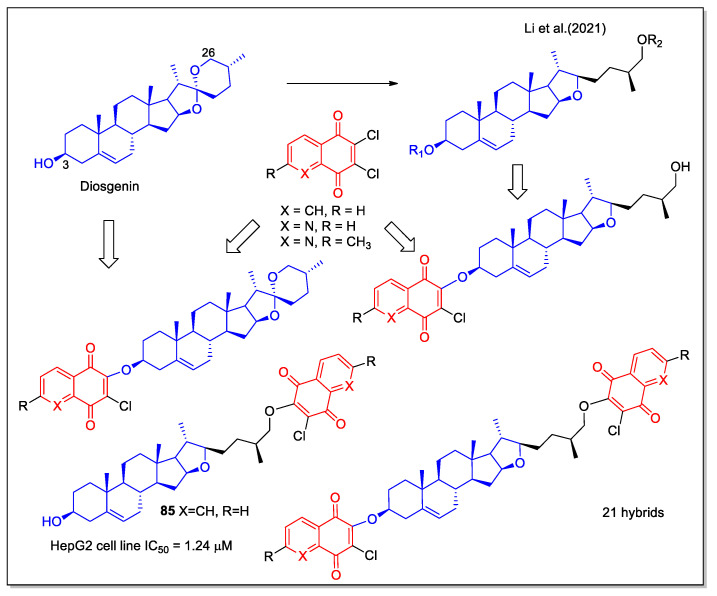
1,4-Naphthoquinone or quinoline 5,8-dione hybrids with diosgenin as reported by Li et al. [72].

**Figure 32 molecules-27-04948-f032:**
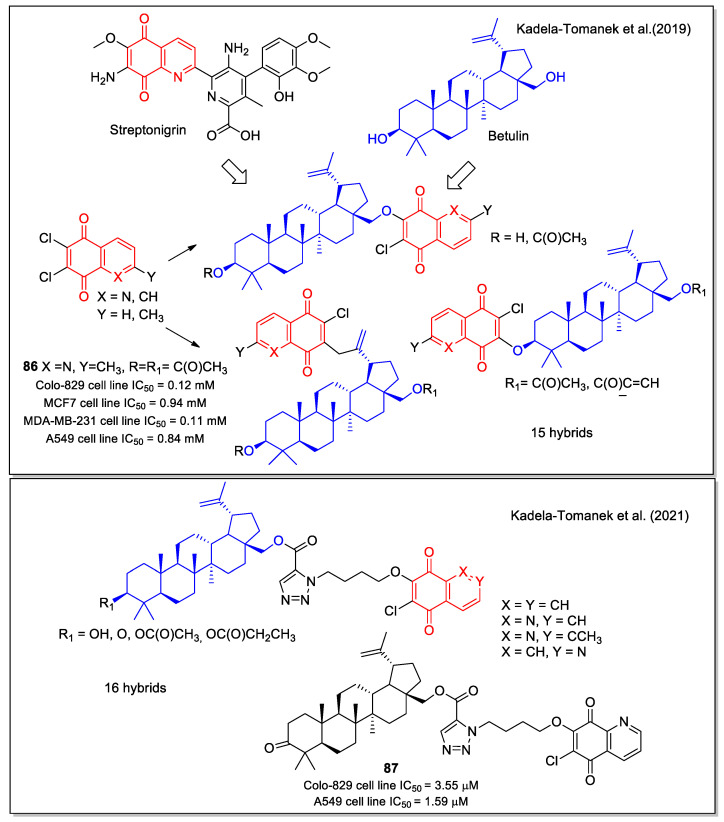
1,4-Naphthoquinone, quinoline 5,8-dione and isoquinoline-5,8-dione scaffolds in hybrids with betulin as reported by Kadela-Tomanek et al. [73,74,75].

**Figure 33 molecules-27-04948-f033:**
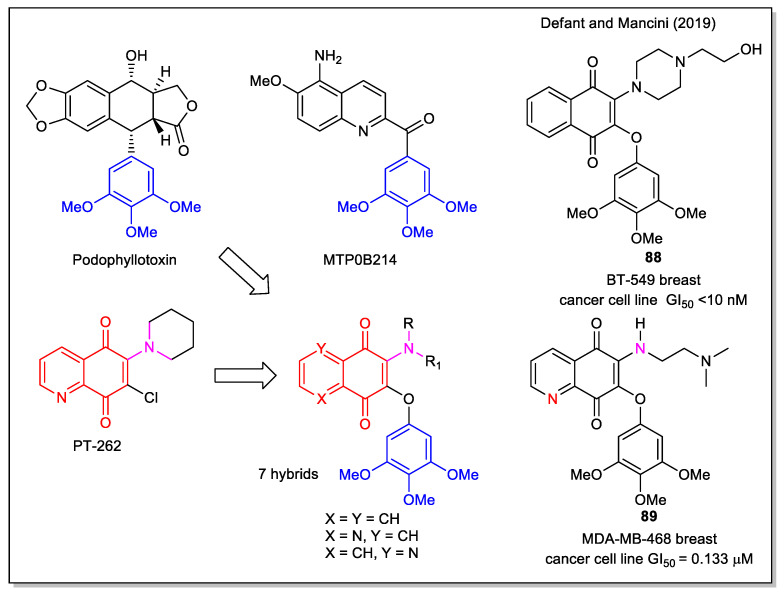
1,4-Naphthoquine and quinolone-5,8-dione scaffolds in alkylamino-trimethoxyl hybrids as reported by Defant and Mancini [76].

**Figure 34 molecules-27-04948-f034:**
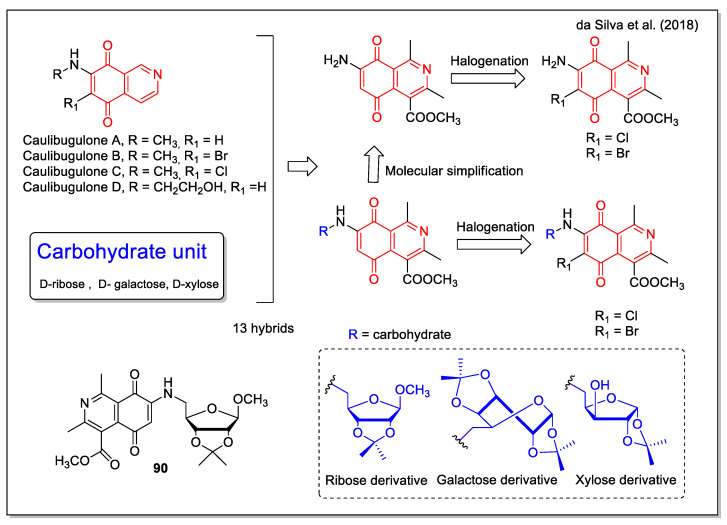
5,8-Isoquinolinedione-carbohydrate hybrids as reported by da Silva et al. [77].

**Figure 35 molecules-27-04948-f035:**
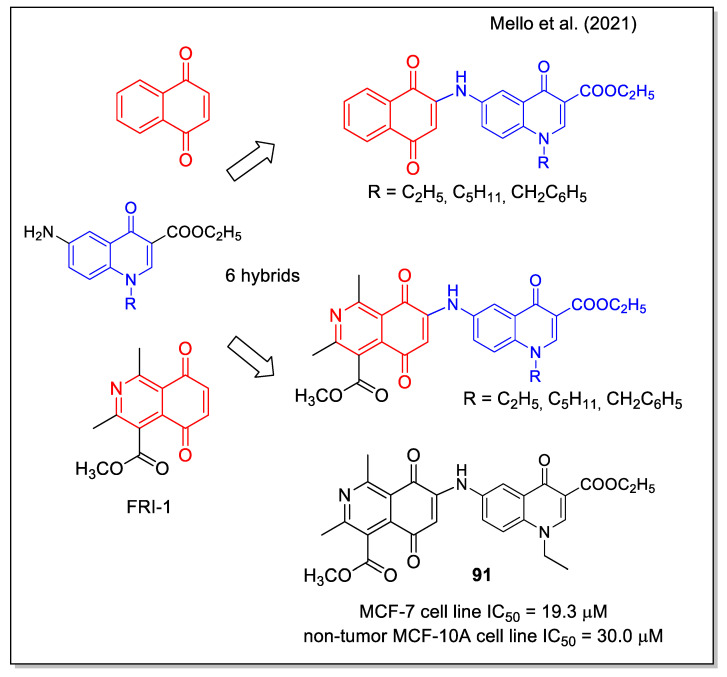
1,4-Naphthoquinone and isoquinolinedione-4-oxoquinoline hybrids as reported by Mello et al. [78].

**Figure 36 molecules-27-04948-f036:**
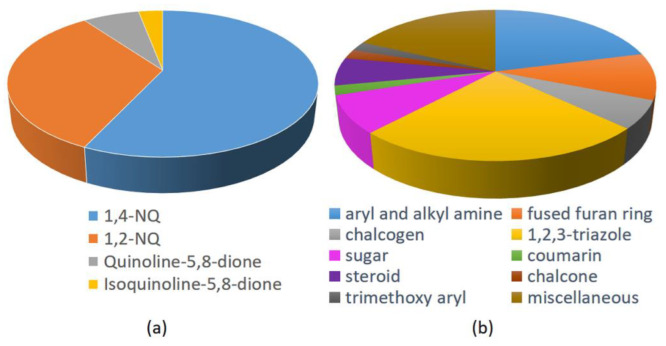
Relative abundance of scaffolds used in the molecular hybridization: (**a**) naphthoquinone types, and (**b**) other fragments, as reported in the papers herein reviewed.

**Table 1 molecules-27-04948-t001:** Summary of data reported in the reviewed works on biological evaluation and computational analyses.

Biological Evaluation			Computational Analyses				Figure	Reference
sub-μM IC_50_	Normal Cell	*In Vivo*	DFT	Docking	Drug Likeness	MD		
			✓				Figure 3	[26,27]
	✓			✓	✓	✓	Figure 3	[28]
			✓	✓			Figure 4	[29]
				✓			Figure 4	[30]
✓	✓						Figure 6	[34]
✓	✓			✓			Figure 7	[37]
✓				✓			Figure 9	[39]
				✓			Figure 10	[41]
	✓						Figure 11	[43]
	✓			✓	✓		Figure 13	[46]
✓				✓			Figure 14	[47]
✓				✓			Figure 17	[52]
				✓			Figure 18	[53]
		✓		✓			Figure 18	[54]
✓				✓			Figure 18	[55]
✓							Figure 19	[56]
✓		✓					Figure 19	[57]
✓	✓						Figure 19	[58]
✓				✓			Figure 20	[59]
	✓						Figure 20	[60]
✓							Figure 21	[61]
	✓						Figure 21	[62]
✓	✓						Figure 22	[63]
✓							Figure 23	[64]
	✓			✓			Figure 23	[65]
✓	✓						Figure 25	[67]
✓	✓						Figure 27	[68]
✓	✓						Figure 28	[69]
✓	✓						Figure 29	[70]
✓	✓						Figure 30	[71]
				✓			Figure 31	[72]
✓					✓		Figure 32	[73,74]
				✓			Figure 32	[75]
✓				✓	✓		Figure 33	[76]
	✓						Figure 34	[77]
	✓						Figure 35	[78]

**Table 2 molecules-27-04948-t002:** Evaluation of the mechanisms of action reported in the reviewed works.

Authors (Year)	Mechanisms of Action
Saluja et al. (2014)	antioxidant activity
Gholampur et al. (2020)	cell cycle arrest in the S phase and potentially apoptosis induction
Rani et al. (2019)	Erβ receptor and protein kinase CK2 potential inhibition by docking
Fiorot et al. (2019)	PI3Kγ and AMPK potential inhibition by docking
Bolognesi et al. (2008)	apoptotic EGFR-mediated intracellular signaling
Gholampur et al. (2019)	cell cycle arrest at G0/G1 phase
Valença et al. (2017)	apparently involves the generation of ROS
Prasad et al.(2018)	induction of cell cycle arrest and apoptosis
da Cruz et al. (2014)	potential releasing of ROS
Alimohammadi et al. (2020)	tyrosine kinase inhibition, BCR-ABL protein, Abl kinase, and T315I Abl mutantas targets by docking
Lin et al. (2018)	cell cycle arrest in G2/M phase as tubulin inhibitor and PDK1 activity inhibition
Bao et al. (2018)	induction of apoptosis in A549 cells in a concentration-dependent manner
Gach et al. (2016)	generation of intracellular ROS
Długosz et al. (2027)	apoptosis induction
Zhou et al. (2018)	potential STAT3 inhibitory activities by docking
Aly et al. (2020)	CDK1/CDC2 phospho-Tyr15 regulation and pre-G1 apoptosis and cell cycle arrest at the G2/M phase
Löcken et al. (2018)	inhibition of STAT3 phosphorylation and generation of ROS in an NQO1-independent manner
Bian et al. (2014)	NQO1-mediated ROS production by experiments and docking
Yu et al. (2021)	NQO1 binding by experiments and docking
Li et al. (2016)	NQO1 substrate and NQO1-mediated ROS production
da Silva Júnior et al. (2009)	no correlation between redox potential and cytotoxicity
Wu et al. (2021)	NQO1 substrate by enzymatic assay; cell cycle arrest in G0/G1 phase, cell apoptosis induction through the mitochondrial pathway, and ROS generation promtion. tumor growth suppression in vivo with no influences on animal body weight.
Martín-Rodríguez et al. (2019)	multi-targeting agent by induction of JNK activity, cell cycle arrest, apoptosis induction and inhibition of the BCR-ABL1/STAT5/c-MYC/PIM-1 signaling pathway
Zhou et al. (2019)	non-intercalative Topo IIα inhibitor, ATP binding site and inductor of apoptosis
Vieira et al. (2015)	ROS generation
Costa et al. (2018)	inhibition of topoisomerase I and IIα but not IIβ by docking
da Silva Júnior et al. (2011)	induction of apoptotic cell death mediated by ROS generation
da Cruz et al. (2016)	bioactivation by NQO1 followed by apoptosis associated with ROS
Gontijo et al. (2017)	ROS generation and cytotoxic action in subcellular lysosomal organelles
Modranka et al. (2021)	cell cycle arrest at the S phase, cell proliferation reduction, DNA damage and apoptosis induction
Li et al. (2021)	activation of the mitochondrial apoptosis pathway (experimental) and NQO1 enzyme interactions by docking
Kadela-Tomanek et al. (2019)	potential NQO1 interaction by docking
Kadela-Tomanek et al. (2021)	substrates for NQO1 experimental and by docking, mitochondrial apoptosis pathway
Defant and Mancini (2019)	potential tubulin, human topoisomerase II and ROCK1 interactions by docking
Mello et al. (2021)	AMPK activation with mTORC1 signaling inhibition, autophagy activation and ER inhibition stress pathway

## Data Availability

Not applicable.

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
