# Peer review of "Hybrid Molecules Containing Naphthoquinone and Quinolinedione Scaffolds as Antineoplastic Agents"

_molecules, 2022, doi:10.3390/molecules27154948_

Round 1
Reviewer 1 Report
The point of this review is good and promising, but some major comments are needed
1- Rewrite abstract to show the aim of the work and present the results in a good way.
2- Introduction is not comprehensive for the topic of the study, please rewrite introduction section to become more show importance of this study and support it with recent references.
3- Design table for each compound includes all recent studies (last three years at least) on biological activity
4- Add new section on mechanism of action of these compounds
Author Response
RESPONSE FROM THE AUTHORS TO REVIEWER’S COMMENTS
We would like to thank the Reviewer for the time and skill he /she devoted to this revision as well as for the constructive comments. Please find below, point-by-point responses to the specific comments and suggestions. We hope that the Reviewer will find our answers satisfactory.
The point of this review is good and promising, but some major comments are needed.
- Rewrite abstract to show the aim of the work and present the results in a good way.
From the authors:
done, modifying especially the central part of the Abstract as: “Naphthoquinones have shown to inhibit cancer cell growth and are considered privileged structures and useful templates in the design of hybrid molecules. The aim of this work is to summarize the current knowledge on antitumor hybrids built using 1,4- and 1,2-naphothoquinone (often considering natural compounds as lawsone, napabucasin, plumbagine, lapachol α-lapachone, and β –lapachone, respectively) and the related quinolinedione scaffolds, reported in the literature up to 2021.”
- Introduction is not comprehensive for the topic of the study, please rewrite introduction section to become more show importance of this study and support it with recent references.
From the authors:
The paragraph 3, which actually reported the status of the art on antitumor hybrids has been inserted in Introduction, leaving as the last part: “This review presents rational approaches ….”. The cited references have been modified accordingly.
- Design table for each compound includes all recent studies (last three years at least) on biological activity
From the authors:
Through a new bibliographic search, we have verified the presence of recent studies on this topic, finding the following reports: a) DFT study on antioxidant action mechanisms of naphthoquinone-urazole hybrids, added as reference 27, on the hybrids by Saluja [26]; b)
the reference by Córdova-Delgado et al. (2021) added as [79] on the mechanism of action of FRI-1as isoquinolinedione scaffold considered by Mello et al. (Figure 35).
- Add new section on mechanism of action of these compounds
From the authors:
In paragraph 3 the discussion on mechanisms of action involved in the antitumor activity of NQs has been improved; moreover, the new table 2 inserted in the paragraph5. Summary remarks, has been specifically added for the compounds reviewed

Reviewer 2 Report
Manuscript is well prepared. The authors have devoted much attention to the relationship with the naphthoquinone system. Unfortunately, the Authors have described only a few compounds with the quinolinodione system. I think that it is worth extending manuscrypt with quinolinodione and isoquinolinedione derivatives from the last 5 years.
However the Author should add more compounds with the quinolinediones moiety.
Author Response
RESPONSE FROM THE AUTHORS TO REVIEWER’S COMMENTS
We would like to thank the Reviewer for the time and skill he /she devoted to this revision as well as for the constructive comments. Please find below, point-by-point responses to the specific comments and suggestions. We hope that the Reviewer will find our answers satisfactory.
Manuscript is well prepared. The authors have devoted much attention to the relationship with the naphthoquinone system. Unfortunately, the Authors have described only a few compounds with the quinolinodione system. I think that it is worth extending manuscrypt with quinolinodione and isoquinolinedione derivatives from the last 5 years.
However the Author should add more compounds with the quinolinediones moiety.
From the authors:
By a careful updated bibliographic research, it has emerged that actually the number of studies reported on these scaffolds in antitumor molecules is considerable, but it is very limited for antitumor hybrids. Anyway, we have found very recent papers that have been now added: a) Modranka et al. 2021[71], figure 30 for quinolinedione scaffold, estending the study already reported by Janeka’s group; b) 5,8-isoquinolinedione-carbohydrate hybrids in figure 34; additionally, hybrids of 5,8-isoquinolinedione by Mello et al.2021 (new figure 35) already present but only for naphthoquinone scaffold.
RESPONSE FROM THE AUTHORS TO REVIEWER’S COMMENTS
We would like to thank the Reviewer for the time and skill he /she devoted to this revision as well as for the constructive comments. Please find below, point-by-point responses to the specific comments and suggestions. We hope that the Reviewer will find our answers satisfactory.
Manuscript is well prepared. The authors have devoted much attention to the relationship with the naphthoquinone system. Unfortunately, the Authors have described only a few compounds with the quinolinodione system. I think that it is worth extending manuscrypt with quinolinodione and isoquinolinedione derivatives from the last 5 years.
However the Author should add more compounds with the quinolinediones moiety.
From the authors:
By a careful updated bibliographic research, it has emerged that actually the number of studies reported on these scaffolds in antitumor molecules is considerable, but it is very limited for antitumor hybrids. Anyway, we have found very recent papers that have been now added: a) Modranka et al. 2021[71], figure 30 for quinolinedione scaffold, estending the study already reported by Janeka’s group; b) 5,8-isoquinolinedione-carbohydrate hybrids in figure 34; additionally, hybrids of 5,8-isoquinolinedione by Mello et al.2021 (new figure 35) already present but only for naphthoquinone scaffold.

Reviewer 3 Report
Ines Mancini et al. describe naphthoquinone (NQ) and quinolinedione (QD) as promising structures for anticancer therapy and analyze discovery of novel modifications lead to interesting applications.
The review is well organized with scheme accurate and with sufficient number of reference (until to 2021).
Add in figure 1 structure vitamin K3 and K2.
To help better readers add in Figure 2 principal targets of NQ and QD as specify in manuscript.
pls add also references :
Redaelli M, Mucignat-Caretta C, Isse AA, Gennaro A, Pezzani R, Pasquale R, Pavan V, Crisma M, Ribaudo G, Zagotto G. New naphthoquinone derivatives against glioma cells. Eur J Med Chem. 2015;96:458-66. doi: 10.1016/j.ejmech.2015.04.039
Pavan V, Ribaudo G, Zorzan M, Redaelli M, Pezzani R, Mucignat-Caretta C, Zagotto G. Antiproliferative activity of Juglone derivatives on rat glioma. Nat Prod Res. 2017 Mar;31(6):632-638. doi: 10.1080/14786419.2016.1214830
Only pls check english language in some parts.
Hence, this reviewer indicate accept this MS for publications after minor revision
Author Response
RESPONSE FROM THE AUTHORS TO REVIEWER’S COMMENTS
We would like to thank the Reviewer for the time and skill he /she devoted to this revision as well as for the constructive comments. Please find below, point-by-point responses to the specific comments and suggestions. We hope that the Reviewer will find our answers satisfactory.
Ines Mancini et al. describe naphthoquinone (NQ) and quinolinedione (QD) as promising structures for anticancer therapy and analyze discovery of novel modifications lead to interesting applications.
The review is well organized with scheme accurate and with sufficient number of reference (until to 2021).
Add in figure 1 structure vitamin K3 and K2.
From the authors: Done
To help better readers add in Figure 2 principal targets of NQ and QD as specify in manuscript.
From the authors: done, additionally details on biological targets and mechanisms of action have been added in the text in paragraph 3 and in the new table 2 (required by another reviewer)
pls add also references :
Redaelli M, Mucignat-Caretta C, Isse AA, Gennaro A, Pezzani R, Pasquale R, Pavan V, Crisma M, Ribaudo G, Zagotto G. New naphthoquinone derivatives against glioma cells. Eur J Med Chem. 2015;96:458-66. doi: 10.1016/j.ejmech.2015.04.039
Pavan V, Ribaudo G, Zorzan M, Redaelli M, Pezzani R, Mucignat-Caretta C, Zagotto G. Antiproliferative activity of Juglone derivatives on rat glioma. Nat Prod Res. 2017 Mar;31(6):632-638. doi: 10.1080/14786419.2016.1214830
From the authors:
We thank the reviewer, but we consider the suggested references inappropriate for the topic of the review. In fact, dozens and dozens of papers have been reported on antitumor naphthoquinones and quinolinediones (including some of ours that we don't mention), but the aim of the review is to consider “hybrid” compounds, as explicitly reported in paragraph 2 "Bibliographic Research Methodology".
However, we have found interesting and now have cited the work by Pavan et al. et al. Nat.Prod. Res. 2017 in paragraph 3 on the discussion of mechanisms of action.
Only pls check english language in some parts.
From the authors: done
Hence, this reviewer indicate accept this MS for publications after minor revision

Round 2
Reviewer 1 Report
Accept